# Multitrait GWAS to connect disease variants and biological mechanisms

**Hanna Julienne**[1]*, **Vincent Laville**[1], **Zachary R. McCaw**[2], **Zihuai He**[3],
**Vincent Guillemot**[1], **Carla Lasry**[1], **Andrey Ziyatdinov**[4], **Cyril Nerin**[1], **Amaury Vaysse**[1],
**Pierre Lechat**[1], **Hervé Ménager**[1], **Wilfried Le Goff**[5], **Marie-Pierre Dube**[6,7],
**Peter Kraft**[2,4], **Iuliana Ionita-Laza**[8], **Bjarni J. Vilhjálmsson**[9,10], **Hugues Aschard**[1,4]*

1 Department of Computational Biology, Institut Pasteur, Paris, France, 2 Department of Biostatistics, Harvard TH Chan School of Public Health, Boston, Massachusetts, United States of America, 3 Department of Neurology and Neurological Sciences, Stanford University School of Medicine, Stanford, California, United States of America, 4 Department of Epidemiology, Harvard TH Chan School of Public Health, Boston, Massachusetts, United States of America, 5 Sorbonne Université, INSERM, Institute of Cardiometabolism and Nutrition (ICAN), UMR_S 1166, Paris, France, 6 Université de Montréal Beaulieu-Saucier Pharmacogenomics Centre, Montreal Heart Institute, Montreal, Canada, 7 Université de Montréal, Faculty of Medicine, Department of medicine, Université de Montréal, Montreal, Canada, 8 Department of Biostatistics, Columbia University, New York, New York, United States of America, 9 National Centre for Register-based Research, Department of Economics and Business Economics, Aarhus University, Aarhus, Denmark, 10 Bioinformatics Research Centre, Aarhus University, Aarhus, Denmark

* hanna.julienne@pasteur.fr (HJ); hugues.aschard@pasteur.fr (HA)

**Data Availability Statement:** All GWAS summary statistics data used in this study are publicly available. Links to each dataset are provided in S1 Table. All other derived data are available in the manuscript or its supporting information files.

## Abstract

Genome-wide association studies (GWASs) have uncovered a wealth of associations between common variants and human phenotypes. Here, we present an integrative analysis of GWAS summary statistics from 36 phenotypes to decipher multitrait genetic architecture and its link with biological mechanisms. Our framework incorporates multitrait association mapping along with an investigation of the breakdown of genetic associations into clusters of variants harboring similar multitrait association profiles. Focusing on two subsets of immunity and metabolism phenotypes, we then demonstrate how genetic variants within clusters can be mapped to biological pathways and disease mechanisms. Finally, for the metabolism set, we investigate the link between gene cluster assignment and the success of drug targets in randomized controlled trials.

## Author summary

Genome-wide association studies (GWAS) established numerous associations between genetic variants and human traits. The anonymized summary of GWAS results is generally made publicly available to the scientific community and can be explored further. Amongst the many possible secondary analyses, one is to study the effect of a genetic variant on several traits (multi-trait GWAS) rather than a unique trait. We compared several tests to conduct multi-trait GWAS on simulated and real data. We detected 322 new associations that were not previously reported by standard univariate GWAS. We then detected clusters of genetic variants having a similar effect across several traits. Focusing

**Funding:** This work has been conducted as part of the INCEPTION program (ANR-16-CONV-0005) (HA). It was also supported by NIH grant R03DE025665 to HA. The funders had no role in study design, data collection and analysis, decision to publish, or preparation of the manuscript.

**Competing interests:** The authors have declared that no competing interests exist.

on two subsets of immunity and metabolism traits, we demonstrate how genetic variants within clusters can be mapped to biological pathways and disease mechanisms. Finally, for the metabolism set, we investigate the link between gene cluster and success of drug targets in randomized controlled trials. We propose this method for improving the functional interpretation of GWAS results.

## Introduction

Genome-wide association studies (GWASs) have identified thousands of significant genetic associations for multiple traits and diseases[1]. Publicly available summary statistics from these GWASs have proven to be invaluable in human genetic studies, enabling a range of secondary analyses without requiring individual-level genotype data and thus, averting major practical and ethical issues[2]. Among others, the estimation of phenotype heritability[3], the derivation of polygenic risk score[4], and the assessment of causal relations between phenotypes[5] are paragons of their critical utility. GWAS summary statistics have also been extremely useful to investigate pleiotropy and genetic correlation between human phenotypes. For example, recent works assessed whether significant loci for a given phenotype are also associated with other traits[6, 7] while others estimated genome-wide[8, 9] and regional[10] genetic correlations among phenotypes. The joint test of multiple traits can also be an efficient way to detect genetic variants missed by univariate screening[11–23], especially those with association patterns that deviate from the observed phenotypic correlation[24–26]. Nevertheless, while simulation studies and examples from real data applications in best case scenarios have confirmed the relevance of multitrait association tests, they have seldom been applied to large-scale datasets.

The application of multitrait association tests to a large heterogeneous set of traits requires overcoming several practical issues including careful preprocessing of individual GWAS summary statistics to avoid statistical artifacts, the estimation of multiple global parameters, and addressing widespread missing summary statistics. We addressed these issues in recent studies, developing the RAISS[27] approach for imputation, and JASS preprocessing and multitrait analysis pipeline software[28]. Nevertheless, the relative performances of existing multitrait tests in real data have not been fully addressed. In brief, two types of methods have been developed, weighted sum of univariate statistics, assuming a specific multivariate genetic effect distribution[12, 14, 15], and an omnibus approach, allowing for one degree-of-freedom per statistic [11, 29], with some approaches using a combination of both[30]. An extensive and fair comparison of these methods is challenging as most face some of the aforementioned practical issues, no readily available implementation[28], and power in real data highly depends on the true (and unknown) multitrait genetic effect distribution. Finally, in addition to the potential ability to detect new variants, there is increasing interest in using GWAS multitrait association to decipher inter- and intra-phenotype genetic architecture[31, 32]. Again, real data applications are scarce, and questions remain regarding the approach to be used, the detectability of the multitrait genetic structure behind genome-wide genetic correlation, their potential matching to biological mechanisms, and their potential clinical utility.

Here, we build on previous works to conduct a large-scale multitrait analysis using GWAS summary statistics from 36 phenotypes categorized into five clinically relevant sets (*Immunity*, *Anthropometry*, *Metabolism*, *Cardiovascular* and *Brain*). We implemented five tests, an omnibus *K* degree freedom test (for *K* GWAS analyzed jointly) similar to[11, 29], and a weighted approach using four alternative weighting schemes, including some partly similar to previously

proposed ones[12, 14]. Comparing the relative performances of these models, we found substantial specificity of the signal identified by each approach, in terms of both association patterns and expressed tissue enrichment. We then used a Gaussian mixture model on the phenotypes with a variant association matrix to identify potential clusters of variants displaying similar genetic multitrait association profiles. In-depth functional analysis of the resulting clusters demonstrates a connection between those profiles and tissue specific expression. This breakdown of multitrait association signals highlighted how the overall genetic correlation between phenotypes can be decomposed into likely distinct genetic pathways. Finally, we used the phenotypes from the *Immunity* and *Metabolism* sets as case studies to demonstrate the matching between the identified profile and known biological pathways. Notably, mapping SNPs with unknown functions to pleiotropy profiles can indicate putative pathways. We conclude by investigating the potential clinical utility of the identified clusters for drug targeting.

## Results

### Multitrait genetic association test

The first step of our study consisted of ensuring the validity of the proposed statistical tests by studying potential bias, and assessing their statistical power under several simulation scenarios. Let us denote the vectors of single nucleotide polymorphism (SNP) *Z*-scores $\mathbf{z} = (z_1, . . ., z_K)$, where *K* is the number of phenotypes (i.e., the number of GWASs analyzed jointly). The first model we used, which we refer to as *sumZ*, assumes that genetic effects across the phenotype analyzed follow a direction specified by a vector of weights $\mathbf{w}$ to form a weighted sum of *Z*-scores. Here we considered four weighting schemes: i) uniform weighting ($sumZ_1$); ii) weighting according to the first principal component of the phenotypic correlation matrix ($sumZ_r$); iii) weighting according to the first principal component of the genetic correlation matrix ($sumZ_g$); and iv) weighting according to the independent component analysis of the *Z*-scores matrix ($sumZ_{ica}$). The second approach, which we refer to as *omnibus*, does not rely on a prior specification on the direction or magnitude of the SNP effect across traits. In brief, it compares, for one SNP, the vector of genetic effects $\mathbf{z}$ with the expected multivariate normal distribution under the null. It is a standard omnibus test based on summary statistics that allows for one degree of freedom *per* outcome (here *per* phenotype). Both approaches (*sumZ* and *omnibus*) rely on a valid estimation of $\Sigma_r$, the variance-covariance matrix between $z_1, . . ., z_K$, under the null hypothesis of no association, and share similarities with previous approaches (see **S1 Text**).

We first performed an in-depth validation of each approach, starting with a series of simulations under an ideal situation, when there were no missing data and the true Z-score covariance matrix, $\Sigma_r$ is known (**S1**–**S3** **Figs**). We further show using both simulated data (**S4 and S5 Figs** and **S1 Text**) and real data from the *UK Biobank* cohort that in the specific case of complete sample overlap between GWASs, the *omnibus* test is asymptotically similar to a multivariate analysis of variance (MANOVA) applied to individual level data (**S6 Fig** and **S1 Text**). The only major potential source of bias we identified is the misspecification of $\Sigma_r$ which can lead to severe type I error inflation (**S7 Fig**). Misspecification can affect all variants, if $\Sigma_r$ is estimated naively using the z-score data as proposed in previous studies (**S8 Fig**). Comparing various approaches, we found that $\Sigma_\mathbf{r}$ can be accurately estimated using *LDscore* regression[9] (**S9 Fig**), and the approach was therefore used to estimate $\Sigma_\mathbf{r}$ along the genome-wide genetic correlation ($\Sigma_\mathbf{g}$) for the 36 phenotypes analyzed (**S2** and **S3** **Tables**). Nevertheless, misspecification can also be SNP-specific when sample size varies across the SNPs analyzed. Per-SNP sample size heterogeneity can induce different proportions of sample overlap and potentially invalidate $\Sigma_r$ for those variants. We illustrate this potential bias by applying *omnibus* tests for the

joint analysis of the four GWASs from the GC consortium[33] (**S10 Fig**). To address this issue, we implemented additional tools to estimate sample size per SNP when missing and subsequently filtered the variants with small sample sizes (**S11** and**S12 Figs** and **S1 Text**). Finally, out of 10 million variants reported for some GWAS, fewer than 1,000 had complete summary statistics for all 36 phenotypes analyzed. While methods exist to impute missing GWAS statistics, we found them to be inaccurate for multitrait analyses and we implemented the RAISS [27] approach we recently developed to ensure valid imputation for our context (**S13 Fig**). All preprocessing steps were recently incorporated into a publicly available toolset[28]. After applying our preprocessing pipeline to all 36 GWAS analyzed, there remained 6,978,319 SNPs with a missing data rate of 45% (59% before imputation).

Next, to illustrate the relative detection ability of each approach, we conducted a series of simulation studies across various scenarios. **S14 Fig** represents the null hypothesis rejection boundaries of each test using a simple example with two traits, when the genetic effect follows a single bivariate normal distribution and when it follows a mixture of two bivariate normal distributions (to reflect our working hypothesis). In this setting, the *omnibus* test displays the largest detection rate. Among the sumZ tests, $sumZ_g$ and $sumZ_{ica}$ show better performances than $sumZ_1$ and $sumZ_r$, especially when the structure of the genetic signal is heterogeneous. We then conducted more extensive simulations focusing on the *omnibus* and $SumZ_g$ tests. **Fig 1** lays out the results of simulations using 10 traits generated under contrasting scenarios. Again, the *omnibus* test shows the best performances, especially when all traits have a high heritability and if the genetic signal is not structured along a specific direction. The statistical power of the *omnibus* test also tends to increase with the sample overlap especially when the environmental correlation was not aligned with the genetic correlation. Interestingly this good performance was also observed for genetically uncorrelated traits when they have a high heritability. In the case of a highly structured genetic effect sampled along a specific direction $SumZ_g$ and the *omnibus* test performed similarly. However, when only one of the 10 traits had a high heritability, the *omnibus* test underperformed compared to the $SumZ_g$, reflecting the cost of additional degrees of freedom in the *omnibus* test.

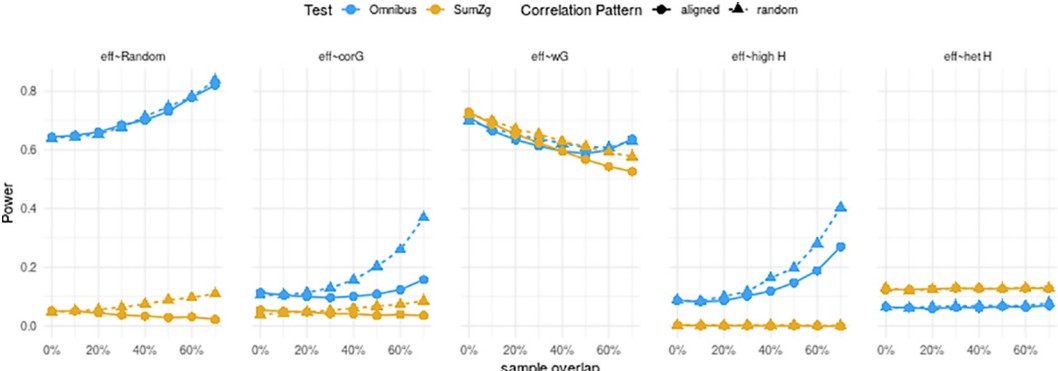

**Fig 1. Statistical Power of the *omnibus* and *sumZ_g* tests under several simulation scenarios.** Power of the *omnibus* and *sumZ_g* tests with respect to sample overlap. Color of the line represents the test. Each panel correspond to a simulation scenario. Point shape indicates if the residual covariance was generated to be partially aligned with the genetic covariance or to be unconstrained (random). A 10 trait Z-score vector was generated as the sum of a genetic effect and a residual effect: $Z = Z_g + Z_{res}$. $Z_{res}$ was sampled from a normal multivariate distribution with covariance matrix terms equal to $\sigma_z = \rho n_s / \sqrt{n_1 n_2}$ where $n_1$ is the sample size of the first study, $n_2$ is the sample size of the second study and $\rho$ is the phenotypic covariance among the $n_s$ overlapping samples. $Z_g$ varied depending on the simulation scenario: (eff ~ Random) $Z_g$ were sampled from a uniform distribution with boundary [-6; 6], (eff ~ corG) $Z_g$ was sampled from a normal multivariate distribution with a random covariance matrix, (eff ~ wG) Zg was sampled along a straight line blurred with a normal noise, (eff ~ high H) $Z_g$ was sampled from a normal multivariate distribution simulating genetically uncorrelated traits with high heritability, (eff ~ het H) $Z_g$ was samples from a normal multivariate distribution simulating genetically uncorrelated trait with only the first having a high heritability.

## Comparison of multitrait association in real data

We analyzed the 36 GWASs of European ancestry (**S1–S3 Tables**) using the aforementioned multitrait approaches applied to seven phenotype sets: five medical-based sets (*Immunity*, *Anthropometry*, *Metabolism*, *Cardiovascular* and *Psychiatric*), a BMI related set including anthropometry traits and lipids (referred further as the *Composite* set), and finally all 36 phenotypes jointly (**Fig 2**). Note that we included bone mineral density traits in the *Immunity* set because an enrichment of BMD genome wide significant loci in immune pathways and immune cell regulatory regions has been previously reported[34, 35]. We derived the overlap of significant loci of the multitrait tests per phenotype set (**S15–S21 Figs**), and after merging all analyses (**Fig 3A**). We applied a Bonferroni correction to the joint tests and used a *p*-value threshold of $10^{-8}$ instead of the standard 5x10$^{-8}$. Univariate phenotype associations were included in the comparison using the minimum of univariate *p*-value across all outcomes (noted $P_{univ}$). Across all phenotype sets, 391 associations were identified by the multitrait tests only, 392 were identified by univariate association tests only, and 1557 were significant for both univariate and multitrait tests (see **Fig 3A**). The largest number of new associations was detected by the *omnibus* test. The performances of the *sumZ* tests varied substantially depending on the phenotype set. For example, the weighting scheme based on phenotypic correlation (*SumZr*), detects slightly more signals than other weights for the *Immunity* set (**S19 Fig**) but

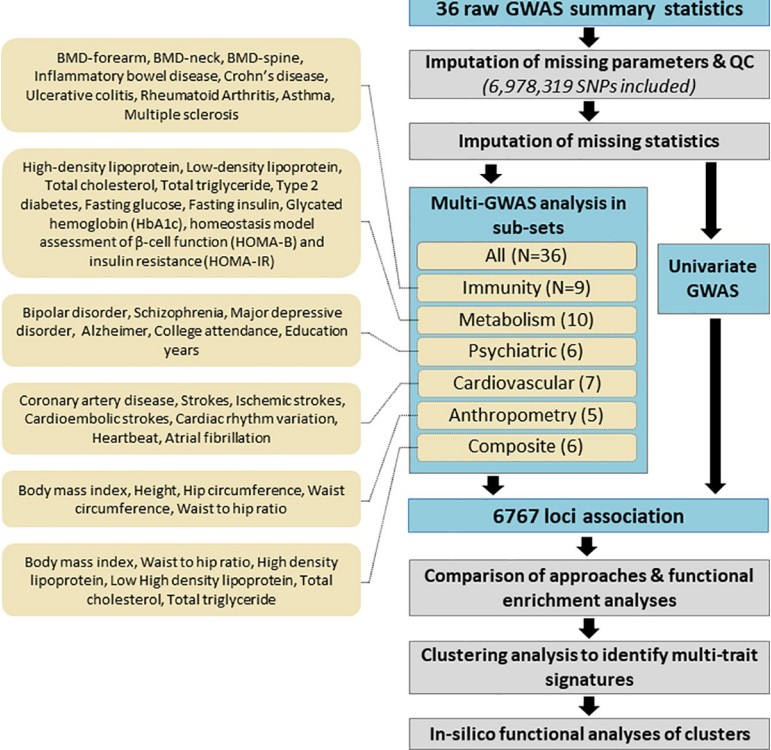

**Fig 2. Analysis overview.** The diagram presents the overall analysis pipeline. A total of 36 GWAS were included covering several common diseases and quantitative traits. All GWAS summary statistics went through extensive pre-processing and quality control filtering, and missing single SNP statistics were imputed when possible. Multitrait approaches were then applied to all clean GWAS data and on each clinically based set (*All*, *Immunity*, *Metabolism*, *Brain*, *Cardiovascular*, *Anthropometry*, and *Composite*). After combining univariate and multivariate results, and merging SNPs within locus, a total of 6,767 associations were identified. After a comparison of results per approach, a clustering analysis was performed for variants within each set. Finally, we performed in-silico functional analysis of the clusters derived in the *Metabolism* set to assess their biological relevance.

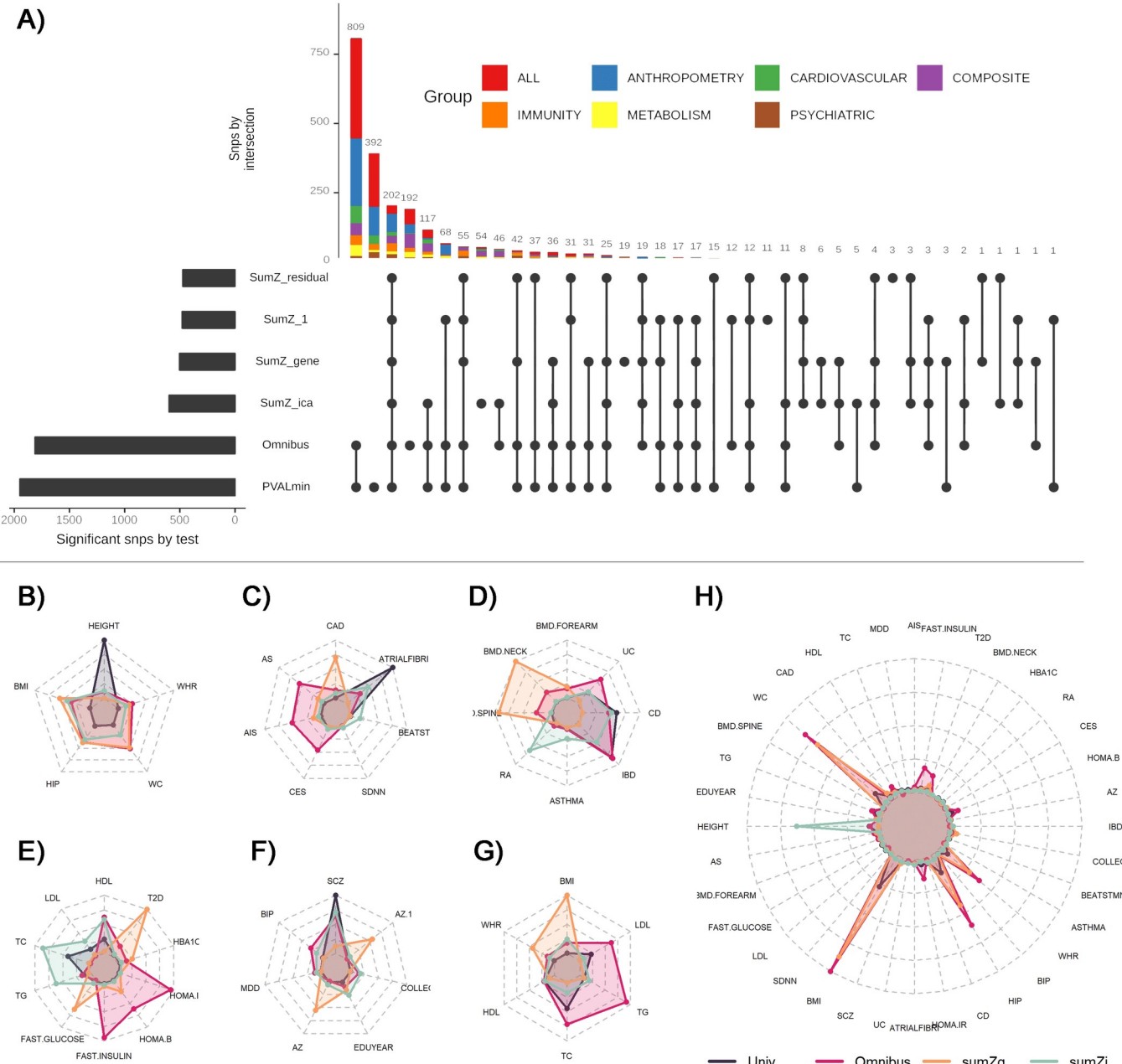

**Fig 3. Multitrait approach comparison.** Panel (**A**) shows independent variants detected across the six approaches: univariate test (*univ*), omnibus test (*omni*), weighted sum of Z-score with uniform weight (*sumZ₁*), weight defined as the loading of the first principal component of the phenotypic correlation (*sumZᵣ*), the genetic correlation (*sumZ_g*), or defined using the loadings of an independent component analysis (*sumZᵢcₐ*). Each line corresponds to a test and each column to a set of significant variants. For each set, the test for which variants are significant are represented with a black dot on the test line. The barplot at the left represents the total number of significant independent signals detected by each approach. The stack bar at the top represents the cardinality of the sets. The next panels show the link between strengths of univariate association signal and the relative performance (i.e. larger power) of the four most tests: *univ*, *omni*, *sumZ_g*, and *sumZᵢcₐ*, for each phenotype set: *anthropometry* (**B**), *cardiovascular* (**C**), *immunity* (**D**), *metabolism* (**E**), *brain* (**F**), *composite* (**G**), and *all phenotypes* (**H**). Within each phenotype set, we split the top associated SNPs per region based on the most significant test, and derived the median chi-squared for each test. The radar plots show the derived median per test and illustrate the strong heterogeneity in patterns identified. For example, out of the 1605 SNPs from the *anthropometry* set, 1235 had stronger signal with *univ* as compared with other tests. The median chi-squares in that group were 49.1, 1.1, 2.0, 1.0, and 0.7 for height, body mass index (BMI), hip circumference (Hip), waist circumference (WC), and waist to hip ratio (WHR). Comparatively, the 267 SNPs harboring a stronger signal with *omnibus*, had median of 6.8, 20.1, 15.9, 11.2, and 7.2 for the same phenotypes.

fewer associations in other phenotype sets (**Fig 3A**). While the *Omnibus* detected the largest number of new associations, the substantial share of signals found by other models suggests that applying several multivariate tests, especially the combination *omnibus, sumZ$_{ica}$, and sumZ$_g$*, could be an interesting solution to maximize detection. Finally, we checked the 392 associations identified by the multitrait test only in these data against previously reported associations from the GWAS catalog[1] for the same phenotypes. Altogether, we report a total of 322 new associations (**S4–S10 Tables**).

To further understand the relative performances of those three tests (*omnibus, sumZ$_{ica}$, sumZ$_g$*) along the univariate test, we explored which multitrait signal was associated with the largest increase in detection per test. To this end, we listed all loci found associated with at least one of the four approaches, and assigned each locus to a test based on the lowest *p*-value. We then derived the median squared z-score by phenotype across the loci assigned to each test. As shown in **Fig 3B–3H**, the median pattern varied substantially across tests and phenotype sets. Higher power for the univariate test was, as expected, observed for strong association signals for a single phenotype, and mostly reflected a very large sample size for that phenotype and/or a strong heritability (e.g. height in the anthropometry set, **Fig 3B**, or atrial fibrillation in the cardiovascular set, **Fig 3C**). A strong association signal for the *omnibus* test was linked to the inclusion of correlated phenotypes and sample overlap, resulting in a high residual covariance ($\Sigma_r$, **S2 Table**). For example, the median squared z-score was elevated for any stroke (AS), any ischemic stroke (AIS) and cardioembolic stroke (CES) in the *Cardiovascular* set. The patterns preferentially detected by the *sumZ$_g$* test are harder to interpret. However, we noticed that *sumZ$_g$* displays a strong signal for SNPs associated with physiologically related traits (e.g., T2D and fasting glucose in the metabolism set, **Fig 3E**, or bone mineral density of neck and spine in the immunity set, **Fig 3D**).

To confirm the relevance of the associations detected by multivariate tests, we also conducted a tissue enrichment analysis to significant variants identified by the multitrait approaches and by the univariate analyses separately (**S11 and S12 Tables**). Overall, univariate variants and multitrait variants harbored a very similar functional enrichment landscape (**S22 Fig**). Most enriched tissues are already known to be involved in the phenotype in question, including liver, fat and pancreas for the *Metabolism* set, immune cell types and thymus for the *Immunity* set, and heart for the *Cardiovascular* set. Our enrichment study also confirmed less obvious observations, which have nevertheless been noted before: the involvement of immunity in brain-related traits (e.g. autisms and schizophrenia)[36, 37] and the overrepresentation of brain tissues in the *Metabolism* set[38, 39].

## Distinct genetic association profiles correspond to distinct genetic correlations

Our comparison of approaches highlights that associated genetic variants display a broad range of multitrait association profiles. We investigated how these profiles can be broken down into groups of homogeneous multivariate genetic effects. This is directly related to the principle of genetic correlation, which quantifies the concordance of genetic effects across traits (e.g. [9]). The difference here, is that genetic correlation captures only the average over the whole genome, and as discussed in recent studies, more localized genetic structures likely exist for many pairs of traits[10]. To detect such a structure, we implemented a multivariate Gaussian mixture model (MGMM)[40] for the identification of clusters among SNPs found associated with at least one approach. We applied MGMM assuming between 2 and 10 clusters and used the BIC and silhouette criteria to determine the most relevant number of clusters. We further bootstrapped the computation of the clustering criteria to ensure the robustness of

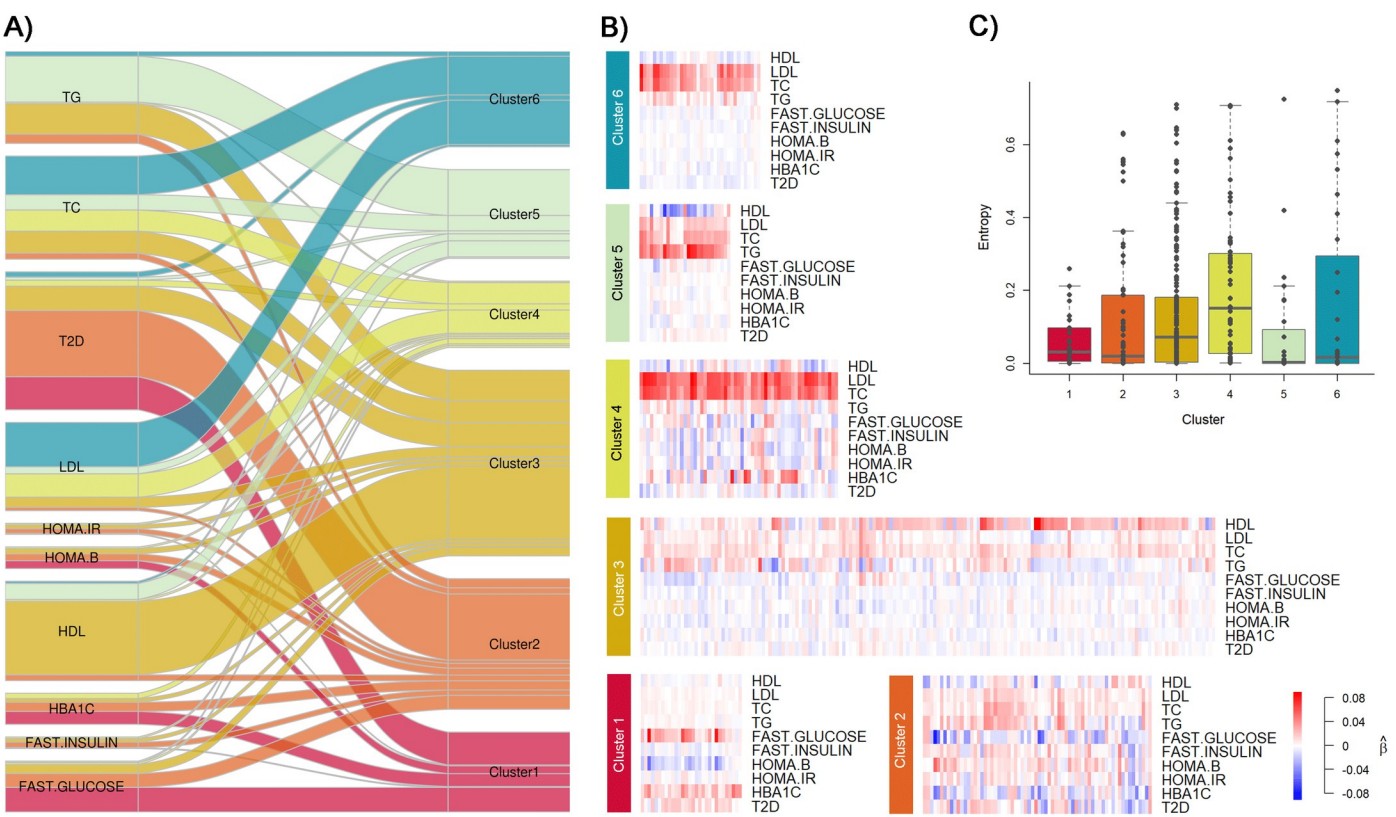

**Fig 4. Multitrait genetic association clusters for the *Metabolism* set.** The panels summarize the clustering of the 392 independent SNPs selected from the *Metabolism* set analysis. The set includes 10 phenotypes: triglyceride (TG), total cholesterol (TC), type 2 diabetes (T2D), low-density lipoprotein cholesterol (LDL-C), high-density lipoprotein cholesterol (HDL-C), glycated hemoglobin (HbA1c), Homeostasis model assessment of β-cell function (HOMA-B), homeostasis model assessment of insulin resistance (HOMA-IR), fasting insulin, and fasting glucose. The alluvial plot in panel **A**) represents the decomposition of univariate genetic association and its rewiring to the six inferred clusters. The flow widths represent the proportion of phenotype's variance explained by the subset of SNPs assigned to each specific cluster, relative to the total genetic variance explained by all 392 SNPs. For example, SNPs from cluster 6 capture approximatively 41.7% and 54.6% of that genetic variance for TC and LDL, respectively. For clarity, flows explaining less than 0.1% of the variance are not represented. Panel **B**) shows the heatmap of normalized beta coefficients per phenotype within each cluster. Each column is a SNP, with blue and red colors indicating negative and positive beta, respectively. Coded alleles have been defined according to the per cluster first principal component. The boxplots in panel **C**) shows the distribution per cluster of SNP's entropy, an indicator of the fitness of the SNP-cluster assignment. SNPs perfectly assigned are expected to have entropy close to zero.

the selection (**S1 Text**). The best suited number of clusters is 6, 8, 8, 9, 3, 2 and 5 for the *Metabolism*, *Immunity*, *Cardiovascular*, *Anthropometry*, *Psychiatric*, *Composite*, and *All* sets, respectively (**S23 Fig**). As illustrated for the *Metabolism* set in **S24 Fig**, adding significant SNPs from the multitrait tests to those identified by the univariate tests enabled us to detect more clusters.

We assessed the uncertainty in cluster assignation by deriving the entropy for each variant (see **S1 Text**). We observed some heterogeneity in the distribution of entropy values across phenotype sets and clusters (**Fig 4C** and **S16** and **S17 Tables**). The difficulty to attribute a cluster for certain variants might be due to the lack of representative cluster (*i.e.* sub-structure not captured in our analysis) or to shared functionalities between clusters that are not modelled in the GMM framework. Because differentiating those two possibilities using the available data would be very challenging, we decided to remove outlier variants ($N = 28$ across all sets) with entropy above 0.75 from further analyses.

The resulting clustering is presented in **Fig 4** for the *Metabolism* set and in **S25-S31 Figs** for the other sets. Each figure presents a heatmap of Z-scores along with an alluvial plot displaying

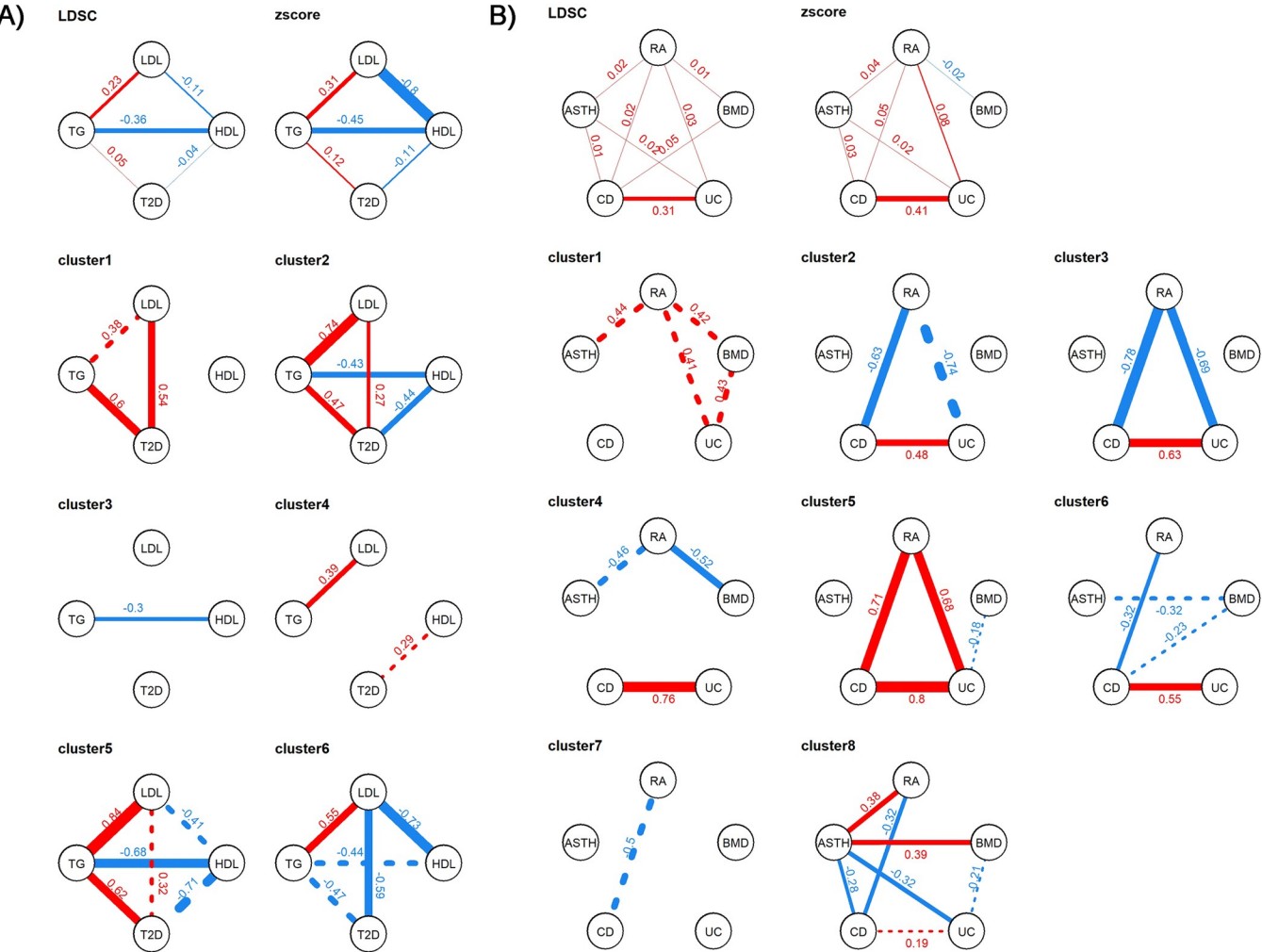

**Fig 5. Heterogeneity of genetic correlation across clusters for the *Metabolism* and *Immunity* sets.** We derived the genome-wide genetic correlation between phenotypes using *LDscore* regression and using Pearson correlation from all SNP Z-scores (top panels), and for SNPs within the identified clusters. Results for the *Metabolism* set are presented in panel (**A**) using only the four key traits, LDL, HDL, Triglyceride (TG) and type 2 diabetes (T2D). Results for the *Immunity* set are presented in the panel (**B**). For clarity only significant correlation are represented. The boldness of the line is proportional to the strength of the genetic correlation. Positive correlations are represented in blue and negative correlations in red. The values of the genetic correlation are indicated by the number next to the trait. Solid lines represent significant correlation (after Bonferroni correction) whereas dashed lines represent correlation significant only before Bonferroni correction. Note that because the clusters are inferred from the multivariate associations, the absolute value of the significance of the correlations is of limited interest. Nevertheless, it provides a useful descriptive statistic to identify the key structures within each cluster.

both the shared explained variance between phenotypes and the proportion of explained variance by clusters for each phenotype. The complete list of SNPs for the *Metabolism* set per cluster is presented in **S16 Table**. The multivariate effects vary substantially from one cluster to another. For instance, in *Metabolism* clusters, SNPs from the cluster 3 display increased HDL-C and decreased triglycerides, while SNPs from cluster 5 are more specific to triglycerides.

The alluvial figures and heatmaps provide an overview of the magnitude of genetic effects from one cluster to another. To further characterize the concordance or discordance of genetic contributions across phenotypes, we computed the pairwise SNP-based genetic correlations for each cluster (see **S1 Text**). **Fig 5** presents those estimates for a subset of phenotypes within the *Metabolism* and *Immunity* phenotype sets. In the *Immunity* set, the correlations between

rheumatoid arthritis (RA), ulcerative colitis (UC) and Crohn's disease (CD) provide a striking illustration of how the genome-wide genetic correlation can be composed of smaller structures. The genome-wide genetic correlations between UC and CD is strong (0.41), but near 0 and not significant for RA (see **S3 Table**). In **Fig 5B**, we noticed a fairly large negative correlation in clusters 2 and 3 between RA both CD and UC whereas, the cluster 5 captures a group of variants displaying a strong positive correlation across the three traits. Similar negligible genome-wide correlations along with opposite genetic correlations across clusters were observed in the *Metabolism* set. For example, variants from cluster 1 display a strong concordant effect between LDL and T2D, but variants from cluster 6 harbor an equally strong negative correlation. **Fig 5** also highlights that significant genome-wide genetic correlations across highly related phenotypes such as UC-CD and LDL-TG are not distributed evenly across variants.

## Biological meaning of genetic clusters

We conducted series of *in silico* functional analyses with the objective of mapping clusters to candidate biological functions. For each phenotype set, we evaluated two types of enrichment: tissue-specific chromatin mark enrichment per cluster (**S13 Table**) and a pathway enrichment framework (**S14 and S15 Tables**) which integrates multiple databases such as Gene Ontology (GO) and KEGG. Here, we focused on the *Immunity* and *Metabolism* sets as a case study. Given a phenotype set, while using the same set of traits for all clusters, we observed large differences in pathways, tissues and cell type enrichments between clusters.

For the *Immunity* set, clusters 1 and 4 predominantly capture genetic effects of bone-mineral density; clusters 2, 3 and 5 affect inflammatory bowel disorder (IBD); and clusters 6, 7 and 8 capture variants with pleiotropic effects on rheumatoid arthritis and IBD (**S27 Fig**). Both enrichment analyses pointed toward an overrepresentation of the immune system with all clusters–even those affecting primarily bone-mineral density–being enriched for at least one immunologic pathway or one immunological tissue. We highlight the top enriched tissues and top pathways in **Table 1.** Concerning pathway enrichment, immune related pathways regulating the shape of the immune response, such as cytokines and the JAK-STAT signaling pathway were recurrent. Interestingly, variants from those clusters map to a distinct set of cytokines and cluster of differentiation genes (e.g., IL4, IL13, IL33 for cluster 1 and IL3, IL5, IL10, IL19, IL20, IL21, IL27 for cluster 5), which suggests that they may impact different components of the immune system. Concerning tissue-specific active chromatin mark enrichment, clusters 2 and 3 contained multiple SNPs enriched primarily in transcriptionally active regions of "primary natural killer cells from peripheral blood" whereas cluster 7 and 8 are enriched for "primary T helper cells." We also observed enrichment in the tissue where immune damage occurred for cluster 5 (colonic mucosa), which highlights the complex interaction between the immune system and the inflamed tissue.

The *Metabolism* set includes several molecular phenotypes, which we expect to be closer to biological mechanisms than some of the macrophenotypes from other sets. Overall, cluster 1 is mostly associated with an increase in fasting glucose and impaired β-cell function; cluster 2 is highly pleiotropic and notably increased the risk of T2D; and clusters 3 to 6 are mostly associated with lipids (**Fig 4**). Accounting for the direction of effects, we also noted that the genetic associations in cluster 5 match the known phenotypic correlation with the inverse relationship between circulating levels of HDL-C with those of LDL-C and, more especially, TGs observed in epidemiological studies[41]. At the tissue level, we observed modest enrichment for adipocytes in clusters 1 and 2 (FDR *p*-value 0.028 and 0.01 respectively, **S13 Table**) and cluster 3 SNPs are up-regulated in the liver (FDR *p*-value 0.005).

**Table 1. Top tissue associations and Immune related Genes by Clusters for the Immunity set.**

| Cl. | #SNP[a] | #gene[b] | Top GTEx Tissue | (q-val) | Top Immunologic pathways | (q-val) | Immunity related genes |
|---|---|---|---|---|---|---|---|
| 1 | 32 | 55 | - | - | http://www.gsea-msigdb.org/gsea/msigdb/geneset_page.jsp?geneSetName=GOMF_CYTOKINE_ACTIVITY&keywords=GOMF_CYTOKINE_ACTIVITY | (1.9 x10$^{-3}$) | IL4, IL13,IL33, STAT6, TNFSF11, TSLP, FAM3C, TNFRSF11B |
| 2 | 40 | 55 | Primary Natural Killer cells from peripheral blood | (6.2x10$^{-5}$) | KEGG_JAK_STAT_SIGNALING_PATHWAY | (1.9 x10$^{-9}$) | IL10, IL12B, IL3, IL4, IL5, IL13, IL19, IL3, IL12RB2, IL23R, CSF2 |
| 3 | 83 | 190 | Primary Natural Killer cells from peripheral blood | (2.8x10$^{-4}$) | KEGG_JAK_STAT_SIGNALING_PATHWAY | (6.2 x10$^{-7}$) | IL3, IL26, IFNG,IL12RB2, IL17REL, IL23R, IFNGR2, CD244, CD274, STAT5A, STAT3, LIF, OSMR, CSF2, CCL13, CCL1, TNFSF15, TNFSF8, JAK2 |
| 4 | 39 | 96 | Bone Marrow Derived Cultured Mesenchymal Stem Cells | (3.5 x10$^{-2}$) | KEGG_JAK_STAT_SIGNALING_PATHWAY | (0.020) | IL2, IL21, IL1R1, IL1RL2, CSF3, STAT3, SPRY1, TSLP |
| 5 | 170 | 430 | Colonic Mucosa | (7.5 x10$^{-5}$) | https://www.gsea-msigdb.org/gsea/msigdb/geneset_page.jsp?geneSetName=IMMUNE_SYSTEM_PROCESS&keywords=IMMUNE_SYSTEM_PROCESS | (1.8 x10$^{-8}$) | IL3, IL5, IL10, IL19, IL20, IL21, IL27, IL12RB2, IL18R1, IL1R2, IL1RL1, IL23R, CD19, CCL2, CCL7, CCL11, NOD2, TNFRSF9, JAK2 |
| 6 | 90 | 198 | - | - | https://www.gsea-msigdb.org/gsea/msigdb/geneset_page.jsp?geneSetName=IMMUNE_RESPONSE&keywords=IMMUNE_RESPONSE | (1.3 x10$^{-7}$) | ILF3, IL12RB2, IL18RAP, IL23R, CD28,CD40, C5, STAT4, STAT1, TYK2 |
| 7 | 20 | 18 | Primary T helper naive cells from peripheral blood | (7.5 x10$^{-5}$) | - | - | |
| 8 | 121 | 59 | Primary T helper memory cells from peripheral blood 2 | (2.5 x10$^{-4}$) | - | | IL6R, TNFAIP3 |

Cluters (Cl.) not mapping to neither tissues nor pathways are indicated by a "-" sign. All reported p-value are FDR corrected.

[a] Count includes only the most associated SNP per region.

[b] Count of genes mapped to SNPs.

As shown in **S14 Table**, each cluster was significantly enriched for a large number of GO terms. We report some specific and illustrative examples: cluster 1 is enriched for the carbohydrate homeostasis set ($q$-value = 2.5 x 10$^{-3}$), cluster 3 is enriched for the reverse cholesterol transport set ($q$-value = 2.8 x 10$^{-13}$), cluster 4 is enriched for the plasma lipoprotein clearance set ($q$-value = 1.7 x 10$^{-5}$), cluster 5 is enriched for the protein lipid complex assembly set (q-value = 1.08x10$^{-9}$) and cluster 6 is enriched for the low density lipoprotein particle remodeling set ($q$-value = 1.07x10$^{-2}$). Cluster 4 also exhibits active chromatin tissue enrichment in immune T cells (q-value = 2.3x10$^{-3}$), highlighting the link between cholesterol and immunity. Indeed, cholesterol and modified forms of cholesterol, such as oxidized cholesterol and cholesterol crystals, promote inflammatory and immune responses through multiple pathways including the activation of the Toll-like receptor (TLR) signaling, the *NLRP3* inflammasome and myelopoiesis[42, 43]. While the promotion of inflammation and immunity is carried by LDL particles, HDL particles were proposed to counteract this effect in part through reverse cholesterol transport[44]. However, cluster 3, which is enriched for reverse cholesterol transport did not exhibit such tissue enrichment in immune T cells, indicating that the link between HDL and immunity may be more complex, as recently pointed out by Madsen et al[45].

## Metabolism pathways and diseases

To provide a perspective on the specificity of genetic variants across clusters and their potential contribution to human diseases, we investigated the lipids from the *Metabolism* set. We first projected each cluster gene onto KEGG pathways. Here, we used only maps corresponding to enriched GO gene sets identified or to tissue identified in the enrichment analysis at the previous stages (**S14** and **S15 Tables**): fat digestion and absorption, cholesterol metabolism, and

PPAR signaling pathways. We constructed a synthesis of these observations on the metabolic map presented in **Fig 6A and 6B**. Genes associated with clusters (**S16 Table**) had functions in agreement with their effects on blood lipid levels: cluster 3 is enriched in genes involved in HDL-C biogenesis and metabolism (*LCAT, ABCA1, SR-B1, CETP, PLTP, LIPG, APOAx and APOCx*), clusters 4 and 6 with genes related to LDL-C clearance (*SORT1, PCSK9, LDLR, LDLRAP1, APOB and APOE*), and cluster 5 to genes related to triglycerides and chylomicron transport (*LPL, APOAx and APOCx*).

We then assessed the effect of variants from each cluster with three diseases known to be associated with serum lipids: coronary artery diseases (CAD), stroke, and obesity (defined as a BMI > 30) (**S19 Table**). Within each cluster, we aligned the SNP alleles with the main trend of the corresponding cluster so that all coded alleles fit the multitrait pattern defined in **Fig 6C** (see **S1 Text**). For example, all SNPs from cluster 5 were recoded to be associated with an increase in TGs, TC and LDL-C and a decrease in HDL-C. We plotted in **Figs 6D** and **S31** the genetic effect of each SNP on the three diseases (using the effect on BMI as a proxy for obesity) after the aforementioned alignment, and we performed a sign test to assess the significance of the observed trend (**S19 Table**). SNPs from several clusters displayed a significant increase in the risk of CAD: cluster 2 ($P = 6.6 \times 10^{-5}$), cluster 4 ($P = 2.9 \times 10^{-2}$), cluster 5 ($P = 3.9 \times 10^{-3}$) and cluster 6 ($P = 2.8 \times 10^{-4}$). SNPs from cluster 2 also displayed a nominally significant increase in the risk of stroke ($P$-value = $1.6 \times 10^{-2}$). Finally, a large fraction of SNPs from cluster 3 had a negative effect on BMI ($P = 6.4 \times 10^{-4}$). Interestingly, several SNPs from this cluster show association with CAD, but with heterogeneous effects–some associated with an increased risk and other associated with a decreased risk–so the absence of a global trend. The associations of clusters 4 and 6 with CAD add to the evidence of a causal effect of LDL-C on CAD[46], which has been established by prospective epidemiological studies[47], Mendelian randomization [48] and randomized clinical trials evaluating the effect of LDL-C reducing therapies[49]. Moreover, the association of cluster 5 with CAD risks corroborates a potential causal role of TGs[5] and remnant cholesterol[50, 51] in CAD. The role of TGs in CAD has also been confirmed by epidemiological studies[52], genome-wide association studies[5], Mendelian randomization studies[53] and randomized controlled trials aiming the lower of TGs[54]. Cluster 3, which is associated with increases in HDL-C, does not have a protective effect on CAD, which is again in agreement with Mendelian randomization analyses reporting no link between HDL and CAD[48, 55]. Finally, the association of cluster 2 with CAD and strokes further supports the potential causal effect of type 2 diabetes on CAD and stroke[56].

As a final exploratory analysis, we reported the cluster and multitrait genetic effect of genes targeted to mitigate hyperlipemia to prevent CAD (**Table 2**). Drug targets corresponding to cluster 3 (*ABCA1, CETP, NR1H3*) did not lead to successful clinical trials, whereas targets (*PCSK9, NPC1L1, APOC3, HMGCR*) in clusters 4, 5 and 6 were mostly successful or promising. The example of the *CETP* gene, that is classified in cluster 3, a cluster not associated with CAD, is of particular interest. *CETP* has been the target of failed clinical trials that attempted to prevent CAD by inhibiting *CETP* and consequently increasing circulating HDL-C[57–59]. Cholesteryl ester transfer protein (*CETP*) promotes the heteroexchange of cholesteryl esters and TGs between HDL-C and APOB-containing lipoproteins connecting HDL-C and TG metabolism[57]. Pharmacological inhibition of *CETP* was motivated by GWAS[60] and prospective cohorts[61] that indicated that *CETP* variants were associated with higher circulating HDL-C levels and lower LDL-C, TGs and CVD risk. However, although all *CETP* inhibitors achieved an effective increase in HDL-C, only *anacetrapib* led to a significantly lower incidence of major coronary events[62] in patients who were receiving statin therapy, an effect that might account for the reduction in *ApoB* (non-HDL-C) rather than the elevation of HDL, as suggested by Mendelian randomization analyses[63]. In addition to these well-known

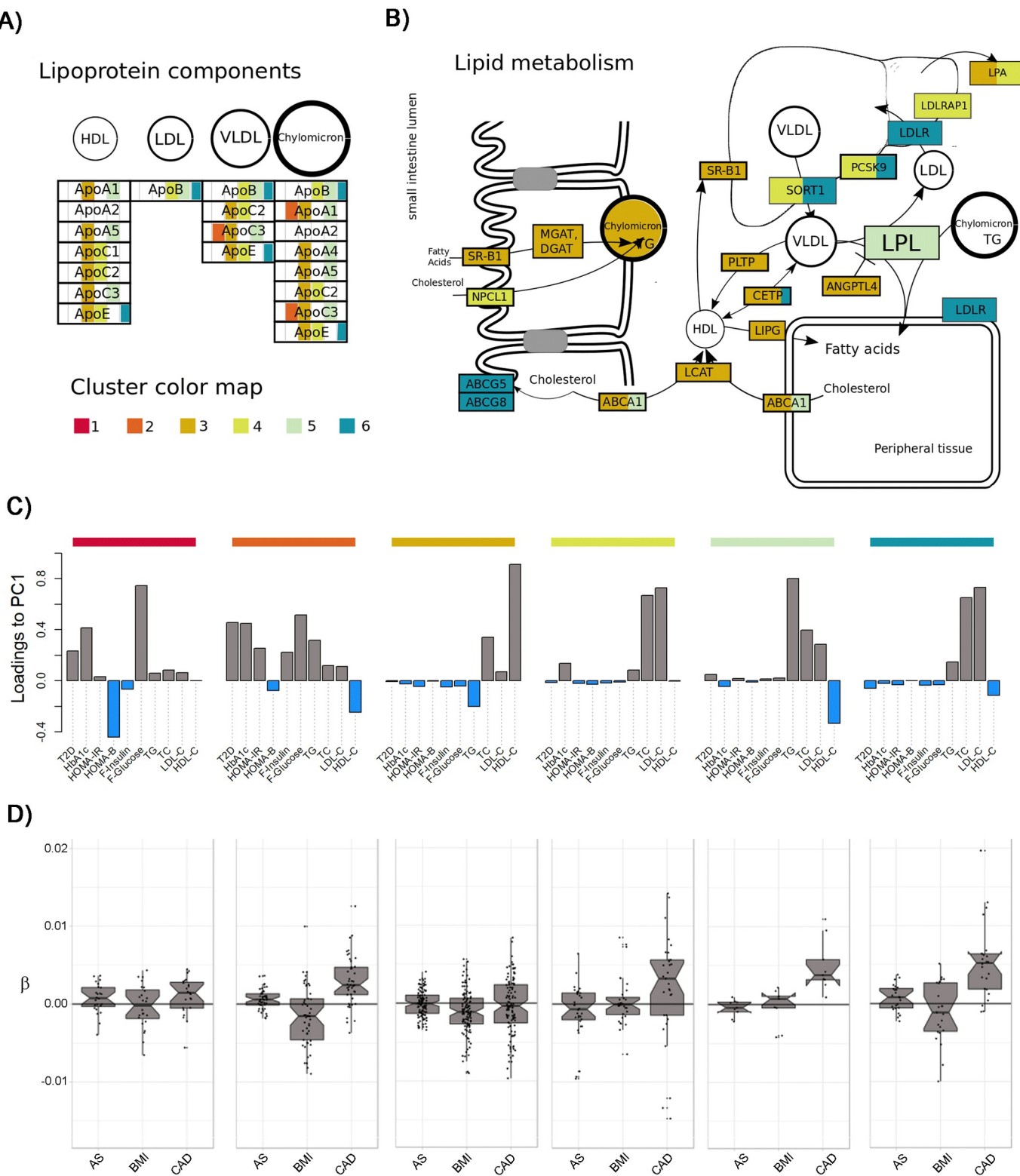

**Fig 6. Mapping clusters to pathways.** We projected cluster's genes from the *Metabolism* phenotype set onto KEGG pathways and reconstructed a synthetic metabolic map. Panel **A**) presents the results for the lipoprotein component and panel **B**) for the lipid component. Gene names are highlighted by the colors of their associated clusters. When a gene is associated to several SNPs belonging to different clusters it is represented with several colors. To improve interpretation, we also present in panel **C**) a proxy for the relative contribution of each phenotype per cluster, defined as the loadings of the first principal component derived from the matrix of Z-score for the subset of SNPs in that cluster. Finally, panel **D**) shows the distribution of standardized beta for association between SNPs from each cluster and three diseases: any stroke (AS), coronary artery disease (CAD), and obesity (using body mass index as a proxy).

**Table 2. Drug target genes and associated SNPs in the metabolism set**

| Target[a] | Drug (phase) | rsID[b] | Clu. | SNP-phenotype association[c] | | | | | | | Comment |
|---|---|---|---|---|---|---|---|---|---|---|---|
| | | | | *HDL* | *LDL* | *TC* | *TG* | *CAD* | *AS* | *BMI* | |
| ABCA1 | Probucol (4) | rs11789603 | 3 | 7.70 | 1.6 | 4.66 | 2.07 | 1.42 | 0.25 | -1.50 | This LDL-c lowering drug was approved but subsequently discontinued because of its lowering effect on HDL-c |
| CETP | Cetrapid (4) | rs12448528 | 3 | 27.79 | -4.61 | 4.96 | -4.60 | 0.25 | -0.28 | 1.21 | Three clinical trials were halted because they showed adverse effect and/or no therapeutic efficacy, except in the case of anacetrapid use for preventing new acute coronary events in high-risk individuals. |
| NR1H3 | HDCA (1) | rs12575609 | 3 | 9.11 | -0.19 | 1.89 | -3.26 | 0.76 | 0.06 | -3.9 | The RCT results were not produced due to AtheroNova Inc. bankruptcy. |
| PCSK9 | alirocumab, Evolocumab (4) | rs7523242 | 4 | -1.16 | 10.49 | 9.28 | 1.92 | 3.25 | 1.03 | -0.29 | Approved second line treatment for high cholesterol individuals whose cholesterol is not controlled by Statin alone. |
| NPC1L1 | Ezetimibe (4) | rs217386 | 4 | -0.80 | 6.60 | 5.96 | 2.44 | 2.19 | 0.86 | -1.47 | Currently used to lower the absorption of cholesterol and is often used in association with statin. |
| APOC3, APOA1 | Volanesorsen (3) | rs1815787 | 5 | -2.09 | 5.45 | 9.41 | 16.60 | 0.39 | 0.26 | 0.614 | A triglyceride-reducing drug currently in phase 3 RCT. |
| HMGCR | Statins (4) | rs59014134 | 6 | 0.79 | 15.79 | 16.06 | 1.34 | 2.01 | -0.36 | -4.59 | The most common cholesterol lowering drugs. |
| APOB | Mipormersen (4) | rs1041968 | 6 | -6.96 | 22.94 | 20.92 | 9.38 | 2.45 | -1.20 | -1.75 | Can be used for risk management in familial hypercholesterolemia but can cause fatty liver disease. |

[a]Note that for probucol, the molecule inhibit ABCA1, but is not specific to ABCA1.

[b]Primary associated SNP and corresponding cluster. But note that for several loci, there is a few other SNPs from other cluster.

[c]Define as the association Z-score for the most associated variant in the gene.

drugs, we provide a systematic listing of potential drug targets by cluster (**S20 Table**) based on the druggable genome database[64].

Altogether, these results suggest that drug development might be more effective by accounting for the gene context, *i.e.*, by selecting candidate genes not from their individual features but based on the disease association trend of genes displaying similar multitrait association profiles. Under this working hypothesis, the proposed inference of genetic functional groups can provide a means to identify those genes and therefore to select potential candidates.

# Discussion

In this study, we conducted multitrait analyses of GWAS summary statistics from 36 human phenotypes combining association tests and clustering to detect the shared and specific genetic substructures underlying those phenotypes and to explore the links between those substructures and biological pathways and diseases. The question of substructures underlying genome-wide genetic correlation has been partially explored in other recent studies[8, 10]. Our work is in agreement with these studies, confirming the presence of regional genetic correlation differences and offering a data-driven approach for identifying primary substructures across millions of possibilities. Nevertheless, an understanding of whether those genetic functional groups are only statistical constructs or correspond to meaningful biological mechanisms is critical. In the latter, it means that a data-driven approach, such as the one proposed in the present study, can be used to dissect the genetic contribution of complex human phenotypes. Here, we used two complementary functional enrichment analyses to map these multitrait association profiles to pathways, and we report a detailed view of these profiles for the *Immunity* and *Metabolism* phenotype sets. In these analyses, the most enriched tissues varied substantially across clusters within each phenotype set, highlighting the potential of such an approach for characterizing distinct genetic mechanisms.

The variability in pleiotropy profiles across SNPs identified in GWASs has been previously discussed. For example, earlier reports[33] on inflammatory diseases have highlighted such patterns, or proposed grouping of SNPs based on the direction of association[65]. However, those studies used only a handful of SNPs identified at the time of publication. Our analysis based on formal clustering and functional enrichment analyses and using the results of GWASs performed in much larger sample size offers a new and much more detailed qualitative perspective on these profiles. More recent publications have also discussed approaches focusing on the characterization of SNPs displaying pleiotropic effects[66], the inference of shared and distinct genetic pathways between related phenotypes[67], and the identification of genetic components linked to disease subtypes[31, 68]. Our approach shares objectives with some of these methods but also has unique features and advantages. Studies based on component decomposition techniques alike principal component analysis[31, 67] yields endotypes of interest from a biological standpoint, but do not provide the SNP-level genetic decomposition that we are addressing. Approaches that rely on individuals' genotypes are limited by the ethical and practical cumbersome aspects tied to this type of data[68], sometimes without increasing statistical power. For example, as demonstrated in this study, the MANOVA derived from individual-level data is equivalent to the proposed *omnibus* test derived using the summary statistics only.

Past studies have shown that sufficiently curated genetic information can enhance the chance of success of clinical trials[69, 70]. We further argue that fine analysis of pleiotropic effects, as performed in the present study, is a very promising path forward to help identify drug targets with a minimal risk of serious side effects. In particular, the picture of the links between coronary artery disease risk and lipid pathways inferred from our analysis is coherent with the state-of-the-art, while providing critical new evidence. While the association of LDL-C and TGs with CAD is largely documented[46, 71], the relation linking HDL-C with CAD is more complex, as both low and high HDL-C levels have been associated with a risk of cardiovascular disease and mortality[72, 73]. Recent studies pointed out that the functionality of HDL rather than the static measure of its circulating cholesterol level accounts for the relationship between HLD-C and CVD and mortality[73, 74], with a potential role of HDL in remnant cholesterol transport. Overall, evidence for the presence or absence of a causal effect between lipid cholesterol measures and CAD as reported by Mendelian randomization analyses should be considered with caution, as lipid traits result from a complex interconnexion of multiple biological pathways. Our analysis suggests that the genetic contribution to the established negative correlation between HDL-C and CAD might be driven only by a subset of genes within a few specific genetic pathways. Under this hypothesis, drugs targeting mechanisms outside these pathways would be ineffective in decreasing CAD risk. Note that such retrospective analysis can be only suggestive of the potential deleterious side effects of drug targets. Nevertheless, the identification of those candidates might limit cost of further analysis using both *in silico* analyses in independent data, and *ex vivo* study to examine the role of the identified genetic variants on additional intermediate molecular phenotypes (gene expression, protein, etc).

A number of further analyses can be conducted based on the results we obtained. First, we focused on a limited number of phenotype sets. Extending analyses to other sets of phenotypes might help refine potential genetic functional groups and better characterize their link to biological mechanisms. To our knowledge, there are no trivial solutions to solve the intrinsic combinatorial issue (i.e., one can build over $6x10^{10}$ sets of phenotypes from 36 GWASs). Additionally, note that we worked with a data freeze dated from December 2018. Hence, at the date of the publication of this paper, newer summary statistics are available for a few traits. We accounted for these new publications when counting newly identified variants by filtering

associations reported in the latest version of the GWAS catalog. Another critical component of our analysis is the methodological choices for clustering. Here, we considered a Gaussian mixture model, mainly to enable missing values, and used BIC and silhouette to decide the optimal number of clusters. Other methods and alternative criteria might result in slightly different clusters. Moreover, we assume that genetic variants belong to distinct clusters, but it is likely that some variants belong to multiple biological pathways. Note that GMM provides posterior probability of cluster assignment and has the potential to explore overlapping clusters, but better approaches might potentially exist to address that specific question. Additionally, our implementation does not automatically address the problem of allele coding (i.e., the choice of the coded allele) inducing, in some cases, symmetric clusters that we had to merge *a posteriori*. Again, alternative approaches might offer the possibility of solving this issue.

To summarize, we ensured the theoretical reliability of a panel of multitrait tests and demonstrated their capacity to detect new associations on diverse sets of traits. Considering independent significant associations, we stratified SNPs in multitrait profiles corresponding to biological pathways. We believe this stratification to be relevant for multiple applications ranging from functional annotation to drug targeting.

## Online methods

### Multivariate association test

Consider a vector $\mathbf{z}$ of $K$ $Z$-scores statistics for a single nucleotide polymorphism (SNP) obtained from standard univariate genome-wide association screenings of $K$ phenotypes. Under the null hypothesis, $\mathbf{z} = (z_1, \ldots, z_K,)$ follows a normal distribution $N(0, \Sigma_\mathbf{r})$, where $\Sigma_\mathbf{r}$ is the residual phenotypic covariance matrix (**S1 Text**), while under the alternative, $\mathbf{z}$ is expected to display additional covariance due to shared genetics (defined by a genetic correlation matrix $\Sigma_g$). We first considered an *Omnibus* test of the vector of Z-scores, which can be performed using the multivariate Wald statistics:

$$T_{omni} = \mathbf{z}^t \Sigma_\mathbf{r}^{-1} \mathbf{z}$$

where $T_{omni}$ follows a chi-square with $K$ degree of freedom (df) under the null hypothesis of no phenotype-genotype association. We also considered a classic weight-based test defined as:

$$T_{sumZ} = \frac{(\mathbf{w}^t \mathbf{z})^2}{\mathbf{w}^t \Sigma_\mathbf{r} \mathbf{w}}$$

where $\mathbf{w}$ is a vector of $K$ weights applied to the $Z$-score. Under the null, $T_{sumZ}$ follows a chi-squared distribution with 1 degree of freedom. Note that this approach shares similarities with both standard fixed effect meta-analysis[14] and with dimensionality reduction methods (e.g. principal component analysis[25]). One can also note that the *Omnibus* statistics can be expressed as a combination of the *sumZ* statistics over all eigenvectors of $\Sigma_r$(**S1 Text**). We note $\mathbf{v}_i$ the i[th] eigen vector of $\Sigma_\mathbf{r}$:

$$T_{omni} = \sum_{i=1}^{K} T_{sumZ} | \mathbf{w} = \mathbf{v}_i$$

We considered four weighting schemes for the *sumZ* tests: (i) in the SumZ$_1$, $\mathbf{w}$ is equal to the unit vector so all traits have the same weight; (ii) in the SumZ$_r$, $\mathbf{w}$ is equal to the first eigen vector of $\Sigma_r$ so its direction represents phenotypic correlation between traits, (iii) in the SumZ$_g$, $\mathbf{w}$ is equal to the first eigen vector to $\Sigma_g$ so its direction represents genetic correlation between traits, (iii) in the SumZ$_{ica}$ $\mathbf{w}$ is computed by applying an Independent component

analysis (ICA) to the complete matrix of Z-score. To compute the weight vector **w** of the Sum-Z$_{ica}$, for a given phenotype set, the genome wide Z-score matrix was extracted and an independent component analysis was performed with the scikit-learn python package. The component yielding the most novel association was selected as loadings. We verified that this selection procedure did not lead to an inflation under the null hypothesis by simulation (see **S2 Fig**).

Performing the omnibus test requires inverting the Z-score covariance matrix $\Sigma_r$. When this matrix does not have a full rank, we use a pseudo inverse of the matrix based on the singular value decomposition (**S1 Text**). Briefly, as $\Sigma_r$ is a variance-covariance matrix, it can be written **PDP$^t$** where $\mathbf{D} = diag((\lambda_k)_{k=1...K})$, $(\lambda_k)_{k=1...K}$ are the eigenvalues of $\Sigma_r$ and **P** is the orthogonal matrix whose columns correspond to the eigenvectors of $\Sigma_r$. If it is not invertible, only $K'$ eigenvalues are different from 0 (where $K'$ denotes the rank of $\Sigma_r$) and an inverse $\boldsymbol{\Sigma}_r^{-1}$ of the matrix can be computed as $\boldsymbol{\Sigma}_r^{-1} = \mathbf{P}_{K'}\mathbf{D}_{K'}^{-1}\mathbf{P}_{K'}^t$, where $\mathbf{D}_{K'}^{-1} = diag((1/\lambda_k)_{k=1...K'})$ and $\mathbf{P}_{K'}$ denotes the $K \times K'$ matrix whose columns are the $K'$ eigenvectors corresponding to the eigenvalues different from 0. Note that the Omnibus statistics computed with $\boldsymbol{\Sigma}_r^{-1}$ follows a $\chi^2$ distribution with $K'$ degree of freedom.

## Robust estimation of Z-score covariance

The validity of the proposed multivariate tests mostly relies on the accurate estimation of $\Sigma_r$. In practice, the covariance between Z-scores from null SNPs from two GWAS will deviate from 0 when there is both sample overlap and correlation among the traits analyzed. When combining results from two independent studies, or when the trait analyzed has negligible correlation, $\Sigma_r$ will be a diagonal matrix, so that the *Omnibus* test can be performed by summing chi-squared statistics for each SNP to form a $K$ degree of freedom chi-square, and the *sumZ* test becomes a standard weighted meta-analysis of fixed effect. Yet, in the large-scale GWAS era, this situation is unlikely as most of the large GWAS are conducted in the consortium setting, where samples likely overlap across multiple GWAS. It follows that $\Sigma_r$ can contain non-zero off-diagonal terms. Under the complete null model, the expected Z-score covariance for null SNPs between two traits equals $\sigma_z = \rho n_s / \sqrt{n_1 n_2}$ where $n_1$ is the sample size of the first study, $n_2$ is the sample size of the second study and $\rho$ is the phenotypic covariance among the $n_s$ overlapping samples (*see* **S1 Text** and e.g. [3, 9]). In some specific cases, one can obtain these parameters directly from the data (e.g. when analyzing multivariate omics data). Conversely, obtaining all four parameters ($\rho$, $n_s$, $n_1$, $n_2$) from consortium GWAS based on dozen or even hundreds of cohorts can be a practically daunting and risky task. Moreover, accurate phenotypic covariance estimation would be particularly challenging when study-specific and trait–specific covariates adjustment has been performed. Recent studies proposed to estimate $\Sigma_r$ using available SNPs from the GWAS in question using all available single SNPs Z-score [75] or using a random subset of pruned variants[14], though some discussed removing GWAS hits[15], focusing on a subset of SNPs in regions less likely to contain causal variants [76], or using tetrachoric estimator[16]. The validity of these estimators mostly relies on the assumption that the vast majority of the SNP effects in the genome are distributed under the null hypothesis. While this is likely to be true in some cases, associated variants can potentially lead to either upward or downward pairwise covariance between Z-scores. Instead, we leverage recent work by Bulik-Sullivan et al[3, 9] that allows for estimation of this covariance (and the diagonal variance terms) under a polygenic model and assuming multivariate normality of effect sizes across traits (*see* **S1 Text**). The estimation of $\Sigma_r$ was performed on Z-scores before the imputation step described in the next section. For a few traits the estimated variance is markedly inferior to 1. As indicated in the LDSC regression method, this phenomenon happens when the original GWAS was corrected with a genomic control factor.

## Simulation studies on test statistical power

To assess the statistical power of the Omnibus test and the SumZ tests, we designed the following simulation scenarios. A 10 trait z-score vector was generated for each of the 5000 causal SNPs as the sum of a genetic effect and a residual effect: $Z = Z_g + Z_{res}$. In the following, random covariance matrices were generated using the *randcorr* R package[77]. $Z_{res}$ was sampled from a normal multivariate distribution with covariance matrix ($\Sigma_r$) terms equal to $\sigma_z = \rho n_s / \sqrt{n_1 n_2}$ where $n_1$ is the sample size of the first study, $n_2$ is the sample size of the second study and $\rho$ is the phenotypic correlation among the $n_s$ overlapping samples. As demonstrated in[9], $\rho = \rho_g + \rho_e$ where $\rho_g$ is the genetic correlation and $\rho_e$ the environmental residual correlation. We generated the residual covariance matrix $\Sigma_r$ according to two scenarios: (random) $\rho_r$ was randomly sampled with the only constraint that $\Sigma_r$ remains a semi definite positive matrix, (aligned) $\rho_r$ was constructed as $\rho_r = 0.4 \times \rho_g + 0.6 \times \mathcal{E}$ with $\mathcal{E}$ sampled at random. $Z_g$ varied depending on the simulation scenario: (eff ~ Random) $Z_g$ were sampled from a uniform distribution with boundary [-6; 6], (eff ~ corG) $Z_g$ was sampled from a normal multivariate distribution with a random covariance matrix, (eff ~ wG) Zg was sampled along a straight line blurred with a normal noise, (eff ~ high H), $Z_g$ was sampled from a normal multivariate distribution simulating genetically uncorrelated traits with high heritability, (eff ~ het H), $Z_g$ was samples from a normal multivariate distribution simulating genetically uncorrelated trait with only the first having a high heritability.

## Data preprocessing: an overview

The analysis of the 36 GWAS required substantial preprocessing, including the inference of several parameters. First, for many publicly available GWAS, sample size per SNP was not readily available and retrospectively collecting this information can be very challenging as it implies requesting this information from each individual cohort. For such a situation, we propose inferring a proxy for missing sample size as $1/(\hat{\sigma}^2_{\beta_G} \sigma^2_G)$, where $\hat{\sigma}^2_{\beta_G}$ is the variance of $\hat{\beta}_G$, the estimated SNP effect, and $\sigma^2_G$ the variance of the SNP, derived from the coded allele frequency which is either provided with the GWAS or extracted from a reference panel (see **S1 Text**). For linear regression this approximate $N\sigma^2_e$, where $N$ is the true sample size and $\sigma^2_e$ is a residual variance of the outcome in the regression model. For logistic regression our estimator is a proxy for the term $Np(1-p)$, where $p$ is the in-sample proportion of cases, and it therefore assumes that the proportion of cases is relatively stable across SNPs with different sample size.

Another challenging issue was the merging of multiple GWAS with different set of assayed SNPs. Indeed, out of 10 million variants reported for some GWAS, fewer than 1,000 had complete summary statistics for all 36 phenotypes analyzed. We performed an imputation of missing $Z$-scores in each study using the RAISS[27] method we recently developed. The approach uses correlation between SNPs to predict $Z$-score at missing SNPs using available ones and achieves a level of imputation accuracy suitable for multitrait analysis (**S1 Text**). Here we used the European panels from the 1,000 Genomes project[78] as a reference for the estimation of the correlation between SNPs. Overall, imputation did not lead to any observable inflation of the *omnibus* statistic (**S13 Fig**). Nevertheless, as a supplementary quality control (QC), we excluded significant SNPs that were not surrounded by SNPs in linkage disequilibrium with significant or near significant $p$-values ($P < 10^{-6}$).

These two parameter inferences were integrated along other preprocessing operations into a pipeline that is fully described here[28]. Given a reference panel with no ambiguous strand, it consists in the following steps (i) Extract, the coded and alternative alleles, signed statistics (regression coefficient or odds ratio), standard error, p-value, and sample size; (ii) Remove all

SNPs that are not in the reference panel; (iii) Derive $Z$-score for each SNP from signed statistics and $p$-value; (iv) Infer sample size when not available; (v) Remove all SNPs whose sample size is less than 70% of the maximum sample size; and (vi) Infer missing $Z$-scores statistics based on the 1K genome reference panel. After applying our preprocessing pipeline to all 36 GWAS analyzed, there remained 6,978,319 SNPs with a missing rate of 45% (59% before imputation).

## Characterization of new loci

To determine new and existing trait-associated loci we used genome regions formed by linkage disequilibrium (LD) blocks as defined in Berisa et al[79] using a reference panel of individuals of European ancestry. It included a total of 1,704 independent regions ranging from 10 kb to 26 Mb in length, with an average size of 1.6 Mb. For each independent LD region, we extracted the minimum $p$-value over all SNPs contained in the region, and a single univariate analysis $p$-value defined as the minimum across all single phenotype GWAS and all SNPs in the region. We consider that a region is newly detected by a multitrait test if the joint analysis $p$-value is genome-wide significant while its univariate p-value is not (joint analysis $p$-value $< 1 \times 10^{-8}$ and univariate $p$-value $> 1 \times 10^{-8}$). We determined SNPs carrying the signal inside significant region with the plink "clump" function using the following parameters:—clump-p1, $10^{-8}$;—clump-r2, 0.2. We kept the lead SNP by clump for further analysis (gene mapping and clustering).

   To report associations exclusively detected in the current report (**S4**–**S10 Tables**), we filtered out association present in the GWAS catalogue[1] at the date of the 14th of September 2020 (univariate $p$-value $> 5 \times 10^{-8}$) for traits corresponding to our phenotype set. The following trait labels were used to retrieve associations: (Metabolism set) 'Fasting blood glucose', 'Triglycerides', 'LDL cholesterol', 'LDL cholesterol levels', 'HDL cholesterol', 'HDL cholesterol levels', 'Total cholesterol levels', 'HOMA-B', 'HOMA-IR', 'Hemoglobin A1c levels', 'Type 2 diabetes'; (Psychiatric set) 'Schizophrenia', 'Bipolar disorder', 'Major depressive disorder', 'Alzheimer's disease', 'Educational attainment'; (Anthropometry set) 'Height', 'Waist circumference', 'Waist-hip ratio', 'Body mass index', 'Hip circumference'; (Immunity set) 'Bone mineral density', 'Rheumatoid arthritis', 'Ulcerative colitis', 'Inflammatory bowel disease', 'Crohn's disease', 'Asthma'; (Cardiovascular set) 'Coronary artery disease', 'Ischemic stroke', 'Large artery stroke', 'Stroke', 'Atrial fibrillation', 'Heart rate', 'Heart rate variability traits'; (Composite set) 'Body mass index', 'Waist-hip ratio', 'Triglycerides', 'LDL cholesterol', 'LDL cholesterol levels', 'HDL cholesterol', 'HDL cholesterol levels', 'Total cholesterol levels'.

## FUN-LDA tissue enrichment

We computed enrichment for SNPs belonging to regions of open chromatin (more likely to contain expressed genes[80, 81]) in specific tissues in three cases: i) when comparing results across phenotype sets, ii) when comparing univariate results, and iii) when comparing results across clusters. For all analyses we used functional annotations on 127 Roadmap tissues and cell lines defined by integrating activating histone marks (H3K4me1, H3K4me3, H3K9ac, and H3K27ac) with a latent Dirichlet allocation model as implemented in FUN-LDA[82]. The enrichment score for a tissue is based on the number significant SNPs compared with the total number of SNPs in open chromatin region (see **S1 Text**). Enrichment results are reported in **S11**–**S13 Tables**.

## Multitrait genetic association clustering and selection of the optimal number of clusters

We performed a clustering of top associated SNPs for each phenotype set using a Gaussian Mixture model (GMM). Briefly, the assumed generative model is as follows. Consider n independent vectors $Z_i = (Z_{i1}, \ldots, Z_{ij}, \ldots, Z_{id})^t$ in $\mathbb{R}^d$, each arising from one of k distinct clusters. Each vector represents one SNPs. The value $Z_{ij}$ is the Z-score of SNP $i$ on trait $j$. Let $I_{ij} = 1$ if the $i^{th}$ observation belongs to cluster $j$, and define the indicator vector $I_i = (I_{i1}, \ldots, I_{ij}, \ldots, I_{ik})^t$. Conditional on membership to the $j^{th}$ cluster, $Z_i$ follows a multivariate normal distribution, with cluster specific mean $\mu_j$ and covariance $\Sigma_j$. Let denote $\pi^j$ the marginal probability of membership to the $j^{th}$ cluster. The observations can be viewed as arising from the following hierarchical model:

$$I_i \sim Multinomial(1, \pi)$$

$$Z_i(I_{ij} = 1) = N(\mu_j, \Sigma_j)$$

One major difficulty in applying the GMM was to deal with incomplete data. Indeed, even after imputation of some missing statistics, our datasets still contained some missing values. To solve the clustering, we implemented the statistical framework described by Ghahramani et al[83] which we recently implemented in a R package MGMM[40]. This method relies on EM optimization techniques enabling the inference of unobserved variables from observed variables and an assumed Gaussian mixture model. In classical GMM, the only variable inferred is the posterior probability cluster membership. In the MGMM algorithm, missing Z-scores are also inferred taking into account the observed Z-scores and the inferred probability to belong to a cluster.

The model gives for each SNP the posterior probabilities to belong to each cluster, and was therefore assigned to its most likely cluster, as long as its entropy was larger than 0.75. For a given variant $SNP_i$, the entropy was derived as follow:

$$S(\text{SNP}_i) = \sum_{j=1}^{k} P(\text{SNP}_i \in cluster_j) \times \log(P(\text{SNP}_i \in cluster_j))$$

where $k$ is the total number of clusters and $P(\text{SNP}_i \in cluster_j)$ is the posterior probability of $SNP_i$ to belong to cluster $j$. The higher the entropy the more the SNP attribution to one cluster is ambiguous. SNPs with an entropy higher than 0.75 were filtered out of the clustering results.

Clustering was performed on all independent significant SNPs. For each SNP, we defined three $p$-values on the phenotypic group traits: the minimum univariate $p$-value ($P_{univ}$), the *Sum-$Z_{ica}$* $p$-value and the *omnibus* $p$-value. All SNPs with at least one of the three $p$-value under $10^{-8}$ were selected for further analysis. For the *Metabolism* univariate clustering, we only considered the univariate $p$-value to perform the selection. We then applied the plink[84] clump function to retrieve practically independent associations using the 0.2 as clump-r2 parameter and $10^{-8}$ as clump-p1 parameter. For each clump we selected a representative SNPs as the one with the smallest $p$-value across the three tests and having more than 60% of its values observed. Note that for a negligible number of occurrences, the representative SNPs has a p-value above $10^{-8}$ (**S15** and **S16** Tables). We applied MGMM within each phenotype set and varied the pre-specified number of clusters between 2 and 10. To select the optimal number of clusters $k$, we performed the clustering 100 times on a random subset of 80% of the SNPs for each $k$. For each resulting clustering we computed the Bayesian Information Criteria and the Silhouette[85] (see **S23** and **S24** **Figs**). Except for the *Metabolism* set, the silhouette appears conservative and the

BIC criterion anticonservative, *i.e.* the latter criteria tends to select a larger number of clusters. We decided to use the following *ad hoc* compounded criterion:

1. If the optimal number of clusters determined by the BIC criteria is higher than the one determined by the silhouette criteria, starting from the silhouette optimal, increase the number of clusters until one of these two conditions is met: 1) adding one cluster significantly decrease the silhouette criterion, 2) the BIC optimal number is reached.

2. In other cases, set the optimal number of clusters to the one determined by silhouette.

## Cluster genetic correlation

We defined pairwise genetic covariance per cluster for as $\rho_{g,cluster=}\beta_1^t\beta_2/M$ where $\beta_1$ and $\beta_2$ are the vector of genetic effects for the pair of phenotypes considered and $M$ is the number of SNPs in the cluster. To estimate properly this quantity from the observed $\hat{\beta}$, we accounted for the bias introduced by sample overlap and phenotypic correlation using the following estimator (see **S1 Text**):

$$\mathbb{E}\left(\boldsymbol{\beta_1}^T\boldsymbol{\beta_2}\right) = \frac{\mathbb{E}(\hat{\boldsymbol{\beta}}_1^T\hat{\boldsymbol{\beta}}_2)}{M} - \frac{\boldsymbol{n_s\rho_Y}}{\boldsymbol{n_1 n_2}}$$

where $\rho_Y$ is the phenotypic covariance, and $n_s$, $n_1$ and $n_2$ are respectively the sample size shared between the two traits, for the trait 1, and for the trait 2. To assess whether the estimated genetic covariances are significantly different from zero, we performed for each pair of phenotypes within each cluster, a *t*-test on the vector of random variables $(X_1, X_2,\ldots,X_M)$, were $X_j = \hat{\beta}_{j,1}\hat{\beta}_{j,2} - \frac{n_s\rho}{n_1 n_2}$ is the contribution of SNP $j$ to the covariance. Note that we used only independent SNPs selected using LD-clumping with squared-correlation parameter equals 0.2.

## Functional enrichment of metabolism clusters

We used FUMA[86] *SNP2GENE* function to associate SNPs with genes based on two criteria, the physical position (in 30kb radius of a protein coding gene) and eQTLs (all significant cis-eQTL from GTEx up to a distance of 1Mb). Note that we restrained the eQTLs to the one that were found in relevant tissue for the Immunity and Metabolism set: immune cells for Immunity and adipose, intestine, liver and brain tissues for Metabolism (see **S1 Data** for complete parameters). After chaining genes to clusters based on SNPs, we performed a functional enrichment for pathways defined in KEGG[87] and GO[88] databases and derived report *p*-values using FUMA *GENE2FUNC* function. Here, cluster's gene were compared against a background of protein coding genes. Finally, we used the R package pathview[89] to project genes onto KEGG pathways maps.

## Disease-clusters association

For the *metabolism* phenotype set, to provide an indicator of the relative contribution of genetic variants to phenotypes in each cluster from the *Metabolism* set, we performed a principal component analysis (PCA) of the SNP-by-phenotype association matrix within each cluster. For this analysis, we used scaled beta coefficients, *i.e.* Z-scores divided by the square root of the phenotype GWAS sample size. To avoid bias due to the arbitrary choice of the coded allele, we randomly shuffled 20 times the coded allele, and repeated the PCA after each shuffling. We report in **Fig 5**, the average of the loadings of the first PC over all shuffling. Note that the first PC only provides the multidimensional direction explaining the largest variance and

therefore do not fully capture the distribution of genetic effect within each cluster. Nevertheless, those first PCs explained a substantial amount of the total variance, equal to 75%, 38%, 53%, 64%, 80% and 93% of the variance in betas for cluster 1 to 6, respectively.

Then, we assessed the association between SNPs within the inferred cluster and three traits (none of which being included in the *Metabolism* set): cardiovascular diseases, any stokes and BMI. SNP alleles were aligned according to the first principal by clusters determined in the last section. We applied a sign test to assess the concordance of the sign of the projection on PC1 and the sign of Z-score for on the three tested additional traits. For this analysis we used more stringent criteria to ensure the SNPs independence. We selected the subset of *Metabolism* SNPs for which linkage disequilibrium does not exceed 0.2 (clump-r2 set to 0.05), which diminishes the number of SNPs considered from 391 to 285. Concerning the association of SNPs to drug target, we associated drug target to a representative SNPs by selecting the SNP with the lowest entropy and having a positive silhouette.

## URL Resources

We developed the following R and python packages to perform our study:
JASS_Preprocessing: https://gitlab.pasteur.fr/statistical-genetics/jass_preprocessing
JASS: https://gitlab.pasteur.fr/statistical-genetics/jass
RAISS: https://gitlab.pasteur.fr/statistical-genetics/raiss
MGMM: https://cran.r-project.org/web/packages/MGMM/index.html
Figs 3–5 have been generated using the following R packages:
https://cran.r-project.org/web/packages/UpSetR/index.html
https://cran.r-project.org/web/packages/radarchart/README.html
https://cran.r-project.org/web/packages/ggalluvial/vignettes/ggalluvial.html
https://cran.r-project.org/web/packages/gplots/index.html
https://cran.r-project.org/web/packages/igraph/index.html

## Supporting information

**S1 Text. Supplementary Note.**
(DOCX)

**S1 Fig. Distribution of the proposed statistics under the null.** We simulated series of 10,000 replicates, each including z-score statistics for 5 (panels a-c) and 100 (panels d-f) phenotypes and 100K SNPs. For each replicate we applied the *Omnibus* test (panels a and d), the *sumZ* test using weights of 1 (panels c and f) and test *sumZ* test using weights equals to loading of the first principal component of the phenotypic correlation matrix. Left panels show the observed chi-square distribution in blue against the expected one in red. Right panels show the corresponding *p*-value histograms.
(TIF)

**S2 Fig. Solving singular covariance matrices.** We simulated series of replicates, each including 20,000 individuals, 10,000 SNPs additively coded with frequencies ranging from 0.01 to 0.99, and 100 phenotypes. The phenotypes were drawn from a multivariate normal distribution with mean 0 and a variance-covariance $\Sigma_r$ of rank 50. For each SNP, we computed the z-score of association with each phenotype using linear regressions and then derived *p*-values from the *Omnibus* test using three strategies to derive the inverse of $\Sigma_r$: i) considering only eigenvalues greater than a specified threshold $\epsilon$ (left column), ii) replacing eigenvalues below a threshold $\epsilon$ by $\epsilon$ (middle column), and iii) adding a small value $\epsilon$ to the diagonal terms of $\Sigma_r$ (right column). For each strategy we considered three different $\epsilon$ values: $10^{-3}$(first line),

$10^{-6}$(second line), and $10^{-9}$(last line). The plots show the histogram of $p$-value distribution.
(TIF)

**S3 Fig. ICA component selection inflation control.** We simulated 1000 z-score vectors under the null hypothesis for each of the 7 GWAS set. The z-score vectors under the null hypothesis follow a multivariate Gaussian of null mean and of covariance given by the intercept of the LDSC regression. The ICA was applied on the 1000 simulated z-scores. For each component, the sumZ test using the component as weight vector. We then selected the component yielding the most association as final component. We computed the p-value for the 1000 point for the optimal component. The y-axis represents the observed p-value quantiles with respect to the theoretical p-value quantiles.
(TIF)

**S4 Fig. Summary statistics versus individual-level data tests under the null.** We simulated replicates of 10,000 predictors for 500 (left column), 5,000 (middle column) and 10,000 individuals (right column). Predictors were drawn from a binomial with $n = 2$ and probability varying in [0.01–0.99] to mimic genotype data, and further normalized to have mean 0 and variance 1. We then simulated a) 5, b) 20 and c) 100 correlated outcomes independent from the genotypes. To assess the impact of non-normal outcomes on the relationship between the summary-statistics based test and the individual-level data test, the outcomes were first drawn from a multivariate normal distribution with pairwise correlation ranging in [-0.7; 0.7] and then transform to non-normal distributions based on their quantiles, so that on average, 33% of them follow a uniform distribution, 33% a Laplace distribution, and 33% an exponential distribution. For each SNP, we first applied a Multivariate ANOVA (MANOVA, blue line). We then conducted association screenings for each single phenotype separately, and applied the proposed *Omnibus* test (purple line) and the Wilk's approximation (orange line) on the resulting summary statistics. The plots show the observed -log10($p$-value) of each test against the expected value under the null.
(TIF)

**S5 Fig. Summary statistics versus individual-level data tests under the alternative.** We simulated replicates of 10,000 predictors for 500 (left column), 5,000 (middle column) and 10,000 individuals (right column). Predictors were drawn from a binomial with $n = 2$ and probability varying in [0.01–0.99] to mimic genotype data, and further normalized to have mean 0 and variance 1. We then simulated a) 5, b) 20 and c) 100 correlated outcomes. We assumed that 10% of the predictors were causal, each causal predictor being randomly chosen and associated with up to 50% outcomes, randomly chosen with equal probability. Effect sizes for each causal predictor were drawn from a normal distribution with mean 0 and variance 0.0025 (i.e. the total phenotypic variance explained by the predictors). To assess the impact of non-normal outcomes on the relationship between the summary-statistics based test and the individual-level data test, the outcomes were first drawn from a multivariate normal distribution with pairwise correlation ranging in [-0.7; 0.7] and then transform to non-normal distributions based on their quantiles, so that on average, 33% of them follow a uniform distribution, 33% a Laplace distribution, and 33% an exponential distribution. For each SNP, we first applied a Multivariate ANOVA (MANOVA, blue line). We then conducted association screenings for each single phenotype separately, and applied the proposed *Omnibus* test (purple line) and the Wilk's approximation (orange line) on the resulting summary statistics. The plots show the observed -log10($p$-value) of each test against the expected value under the null.
(TIF)

**S6 Fig. Validation of the summary-statistics approach in UK Biobank individuals.** Multi-trait GWAS was performed in real data from the UK Biobank using five traits, 336,347 unrelated individuals, and 14,718 genotyped SNPs on Chromosome 20. We used two methods: MANOVA as implemented in PLINK, and the *Omnibus* test from JASS. The plot shows the -log10($p$-values) of the Omnibus approach as a function of the -log10($p$-values) derived with the MANOVA.
(TIF)

**S7 Fig. Impact of covariance bias on multivariate tests.** We simulated three correlated vectors of z-scores for 10,000 SNPs. We derived the multivariate test for each SNP using either the true covariance matrix ($\Sigma_{\text{true}}$), an upward-bias covariance matrix ($\Sigma_{\text{upBias}}$, left panels) or a downward-bias covariance matrix ($\Sigma_{\text{downBias}}$, right panels). For both scenarios we derived the *Omnibus* multivariate test using either the true or biased covariance matrix. Invalidity of the test based on the biased covariance matrix is illustrated in the resulting QQplots and genomic inflation factor $\lambda_{GC}$ (bottom panels).
(TIF)

**S8 Fig. Impact of causal variants on the covariance estimation.** We simulated series of correlated z-scores for 100,000 SNPs from two outcomes $Y_1$ and $Y_2$. For each simulation we generated a matrix of true genetic effect $\mathbf{b} = (\boldsymbol{\beta_1}\, \boldsymbol{\beta_2})$ of $m$ standardized and independent genotypes for two phenotypes $Y_1$ and $Y_2$ from a multivariate normal with means of 0 variance $h_1^2/m$, and $h_2^2/m$, respectively, and covariance $\sigma_g$, where $h_1^2 = 0.3$ and $h_2^2 = 0.6$ are the heritability of $Y_1$ and $Y_2$. We then generated $\hat{\mathbf{b}}$ defined as $\hat{\mathbf{b}} = \mathbf{b} + \boldsymbol{\varepsilon}$ where $\boldsymbol{\varepsilon}$ was also drawn from a multivariate normal with means 0, variance $1/N_1$ and $1/N_2$, respectively, and covariance $r_e N_s/\sqrt{N_1 N_2}$, where $N_1$, $N_2 = N_1/2$ and $N_s = N_2/2$ are the sample sizes for $Y_1$ and $Y_2$, and the number of shared samples, respectively, and $r_e = 0.4$ is the correlation between $Y_1$ and $Y_2$ across overlapping samples. Finally, we derived the expected z-score for each genotype $\mathbf{z} = (\hat{\boldsymbol{\beta}}_1\sqrt{N_1}\ \ \hat{\boldsymbol{\beta}}_2\sqrt{N_2}) = (z_1\ \ z_2)$, and $\sigma_z = cov(z_1, z_2)$, the covariance between $z_1$ and $z_2$. The left panel (a) show the covariance between z-score for null variants in red, and the observed covariance between all z-scores except those harboring a $p$-value below a given threshold (0 (i.e. no SNP removed), 5x10$^{-8}$, 5x10$^{-5}$, 5x10$^{-4}$, 5x10$^{-3}$, 1x10$^{-2}$, and 5x10$^{-8}$). The right panel (b) shows the same results while assuming only 10% of the variants are causal, while all remaining have $\mathbf{b} = \mathbf{0}$.
(TIF)

**S9 Fig. Validation of the LDSC covariance estimate accuracy in UK Biobank.** We compared LDSC estimates of covariance between summary statistics against its expected value, which equals $\rho N_s/\sqrt{N_1 N_2}$, where $\rho$ is the phenotypic correlation among overlapping sample and $N_s$, $N_1$ and $N_2$ are the sample overlap, the sample size for phenotype 1 and the sample size for phenotype 2, respectively. We used individual-level data for five anthropometric traits and 619,017 SNPS measured in 336,347 individuals from the UK Biobank cohort. We first considered scenarios with complete sample overlap (i.e. $N_s = N_1 = N_2$, panels a-d), so that LDSC estimate is expected to equal the phenotypic correlation. We compared estimates while using 100% (a), 50% (b), 10% (c) and 1% (ds) of the total sample size respectively. The lower matrix triangle in pale red are estimated phenotypic correlation from individual-level data, while the upper triangle in pale blue shades are estimated correlation derived using the LDSC. We then considered scenario where sample overlap is only partial by sub-sampling individuals only for BMI, using 100% (e),50% (f), 10% (g) and 1% (h) of the total sample for that phenotype. The first row, in red, is the expected GWAS covariance knowing $\rho$, $N_s$, $N_1$ and $N_2$. The second row,

in blue, is the estimate derived using LDSC.
(TIF)

**S10 Fig. Bias due to sample size heterogeneity in the GLG consortium GWAS data.** We performed the *Omnibus* test to the four traits from the GLG consortium: high density lipoprotein (HDL), low density lipoprotein (LDL), total cholesterol (TC), and total triglyceride (TG) using all available SNP with complete summary statistics. We plotted the median chi-squared of the resulting test across bin of SNPs defined based the per-SNP standard deviation in sample size across the four traits ($\sigma_{\text{sample size}}$) for SNPs with a minor allele frequency (MAF) below 5% (a), and above 5% (b). The red dashed line indicates the expected 4 degree of freedom chi-square under the null. The shade of blue is proportional to the number of SNPs in each bin.
(TIF)

**S11 Fig. Impact of allele frequency error on sample size inference.** We simulated 1,000 replicates including 20,000 individuals were a phenotype *Y* is simulated independently of a genotype *G* with frequency randomly sampled in [0.001, 0.999]. For each replicate we tested for association between *G* and *Y* and we inferred *w* the weights, which equals the sample size times a constant, using the standard error of the effect estimate and either the in-sample allele frequency or the in-sample allele plus a noise term sample from a uniform with min and max in [0.01, 0.05, 0.10]. When allele frequency plus noise was larger or smaller than 0 or 1, the value was set to 0.001 and 0.999, respectively. Upper and lower panels show the inferred sample size as a function of the true allele frequency when using the identity and logit link functions for modelling and testing for association, respectively. Note that for the later, for each replicate, we simulated 50,000 individuals and considered a disease prevalence of 25%. We then randomly sampled 10,000 cases and 10,000 controls to form replicates of 20,000 individuals. Also, for the sake of comparison, we scaled *w* by a constant in the logit model so that the target is the true sample size.
(TIF)

**S12 Fig. Impact of case/control ratio misspecification on sample size inference.** We simulated 1,000 replicates including 50,000 individuals. For each individual we generated a disease status assuming a prevalence of 25% and an independent genotype *G* with frequency randomly sampled in [0.001, 0.999]. For each replicate we randomly sampled 5,000 cases and 20,000 controls and tested for association between *G* and *Y* using these samples and after sub-sampling cases (i.e. using a sub-sample of the 5,000 cases and the 20,000 controls, top panel) or controls (i.e. using the 5,000 cases and a sub-sample of the 20,000 controls, bottom panel), in order to mimic situation where either cases or controls would be missing for some SNPs. For each experiment we inferred $W_{log} = Np(1-p)$, where $N$ is the sample size and $p$ is the case-control ratio, using the standard error of the effect estimate and the in-sample allele frequency. In both situations (sub-sampling cases or sub-sampling controls), $W_{log}$ is decreasing with increasing percentage of missingness.
(TIF)

**S13 Fig. Effect of imputation of missing z-scores on the Omnibus test statistic.** Each line corresponds to a GWAS set. The left column is the empirical quantile of the *Omnibus* statistic versus the theoretical quantile before imputation. The middle column is the same after imputation. The right column is the empirical quantile of the *Omnibus* statistic before imputation versus the same quantity after imputation.
(TIF)

**S14 Fig. Bivariate Significance boundary for each test.** We simulated series of $N$ = 100K individuals with two correlated outcomes $Y_1$ and $Y_2$ as a function of $m$ = 1000 independent SNPs. The genetic effect of the $m$ variants on $Y_1$ and $Y_1$, denoted $\mathbf{b} = (\boldsymbol{\beta_1}\ \boldsymbol{\beta_2})$, were drawn from a bivariate normal with means 0, variance $h_1^2/m$, and $h_2^2/m$, and correlation $\boldsymbol{\sigma_g}$, where $h_1^2 = 0.4$, $h_2^2 = 0.25$, and $\sigma_g = 0.8$. The vectors of residual $\boldsymbol{\varepsilon} = (\boldsymbol{\varepsilon_1}, \boldsymbol{\varepsilon_2})$ were drawn from a multivariate normal with means 0, variance $1 - h_1^2$ and $1 - h_2^2$, respectively, and covariance $r_e = 0.5$. For each simulation, we derived the z-score for each of the $m$ genotypes $\mathbf{z} = (\hat{\boldsymbol{\beta}}_1\sqrt{N}\ \ \hat{\boldsymbol{\beta}}_2\sqrt{N}) = (z_1\ \ z_2)$ and plotted $z_1$ as a function of $z_2$. We applied the five tests: *Omnibus*, $sumZ_r$, $sumZ_g$, $sumZ_{ica}$ and $sumZ_1$ and highlighted for each of them whether variants had significant $p$-value (red, p<5x10$^{-5}$) or not (blue, p>5x10$^{-5}$). In the results from the upper panels, $\mathbf{b}$ was drawn from a bivariate normal distribution with the $\sigma_g$ parameter was set to 0.24. For the lower panel, $\mathbf{b}$ was drawn from a mixture of two normal distributions both centered on zero and both with variance equal to $h_1^2/m$, and $h_2^2/m$. However, the covariance $\sigma_g$ was set to 0.8 for the first one and -0.65 for the second one.
(TIF)

**S15 Fig. Signal comparison for all phenotypes.** The upper panel shows independent variants detected across phenotype groups and across approaches represented as an *UpSetR* visualization. Matrix lines correspond to a test, each column to a set of significant variants. For each set, the test for which variants are significant are represented with a black dot on the test line. The barplot on the left of the matrix represents the number of significant independent signals detected by each approach. The barplot on the top of the matrix represents the cardinality of the sets. The sets are ordered by cardinality from the largest to the leftmost to the smallest to the rightmost. The bottom panels show quadrant plots, i.e. the -log10($p$-value) for the most significant SNP per region for the *Omnibus* test as a function of the -log10($p$-value) for the most significant SNP per region across all univariate GWAS. Complete results are presented in the left panel, and a zoom around the genome-wide significance threshold is presented on the right panel.
(TIF)

**S16 Fig. Signal comparison for anthropometric traits.** The upper panel shows independent variants detected across phenotype groups and across approaches represented as an *UpSetR* visualization. Matrix lines correspond to a test, each column to a set of significant variants. For each set, the test for which variants are significant are represented with a black dot on the test line. The barplot on the left of the matrix represents the number of significant independent signals detected by each approach. The barplot on the top of the matrix represents the cardinality of the sets. The sets are ordered by cardinality from the largest to the leftmost to the smallest to the rightmost. The bottom panels show quadrant plots, i.e. the -log10($p$-value) for the most significant SNP per region for the *Omnibus* test as a function of the -log10($p$-value) for the most significant SNP per region across all univariate GWAS. Complete results are presented in the left panel, and a zoom around the genome-wide significance threshold is presented on the right panel.
(TIF)

**S17 Fig. Signal comparison for cardiovascular phenotypes.** The upper panel shows independent variants detected across phenotype groups and across approaches represented as an *UpSetR* visualization. Matrix lines correspond to a test, each column to a set of significant variants. For each set, the test for which variants are significant are represented with a black dot on the test line. The barplot on the left of the matrix represents the number of significant

independent signals detected by each approach. The barplot on the top of the matrix represents the cardinality of the sets. The sets are ordered by cardinality from the largest to the leftmost to the smallest to the rightmost. The bottom panels show quadrant plots, i.e. the -log10 (*p*-value) for the most significant SNP per region for the *Omnibus* test as a function of the -log10(*p*-value) for the most significant SNP per region across all univariate GWAS. Complete results are presented in the left panel, and a zoom around the genome-wide significance threshold is presented on the right panel.
(TIF)

**S18 Fig. Signal comparison for composite phenotypes set.** The upper panel shows independent variants detected across phenotype groups and across approaches represented as an *UpSetR* visualization. Matrix lines correspond to a test, each column to a set of significant variants. For each set, the test for which variants are significant are represented with a black dot on the test line. The barplot on the left of the matrix represents the number of significant independent signals detected by each approach. The barplot on the top of the matrix represents the cardinality of the sets. The sets are ordered by cardinality from the largest to the leftmost to the smallest to the rightmost. The bottom panels show quadrant plots, i.e. the -log10(*p*-value) for the most significant SNP per region for the *Omnibus* test as a function of the -log10(*p*-value) for the most significant SNP per region across all univariate GWAS. Complete results are presented in the left panel, and a zoom around the genome-wide significance threshold is presented on the right panel.
(TIF)

**S19 Fig. Signal comparison for immunity phenotypes.** The upper panel shows independent variants detected across phenotype groups and across approaches represented as an *UpSetR* visualization. Matrix lines correspond to a test, each column to a set of significant variants. For each set, the test for which variants are significant are represented with a black dot on the test line. The barplot on the left of the matrix represents the number of significant independent signals detected by each approach. The barplot on the top of the matrix represents the cardinality of the sets. The sets are ordered by cardinality from the largest to the leftmost to the smallest to the rightmost. The bottom panels show quadrant plots, i.e. the -log10(*p*-value) for the most significant SNP per region for the *Omnibus* test as a function of the -log10(*p*-value) for the most significant SNP per region across all univariate GWAS. Complete results are presented in the left panel, and a zoom around the genome-wide significance threshold is presented on the right panel.
(TIF)

**S20 Fig. Signal comparison for metabolism phenotypes.** The upper panel shows independent variants detected across phenotype groups and across approaches represented as an *UpSetR* visualization. Matrix lines correspond to a test, each column to a set of significant variants. For each set, the test for which variants are significant are represented with a black dot on the test line. The barplot on the left of the matrix represents the number of significant independent signals detected by each approach. The barplot on the top of the matrix represents the cardinality of the sets. The sets are ordered by cardinality from the largest to the leftmost to the smallest to the rightmost. The bottom panels show quadrant plots, i.e. the -log10(*p*-value) for the most significant SNP per region for the *Omnibus* test as a function of the -log10(*p*-value) for the most significant SNP per region across all univariate GWAS. Complete results are presented in the left panel, and a zoom around the genome-wide significance threshold is presented on the right panel.
(TIF)

**S21 Fig. Signal comparison for psychiatric phenotypes.** The upper panel shows independent variants detected across phenotype groups and across approaches represented as an *UpSetR* visualization. Matrix lines correspond to a test, each column to a set of significant variants. For each set, the test for which variants are significant are represented with a black dot on the test line. The barplot on the left of the matrix represents the number of significant independent signals detected by each approach. The barplot on the top of the matrix represents the cardinality of the sets. The sets are ordered by cardinality from the largest to the leftmost to the smallest to the rightmost. The bottom panels show quadrant plots, i.e. the -log10($p$-value) for the most significant SNP per region for the *Omnibus* test as a function of the -log10($p$-value) for the most significant SNP per region across all univariate GWAS. Complete results are presented in the left panel, and a zoom around the genome-wide significance threshold is presented on the right panel.
(TIF)

**S22 Fig. Proportion of tissue type enriched by phenotype group.** The union of enriched tissue by phenotype set were mapped to their larger anatomical category (Tissue type) in GTEx. To simplify the visualization, anatomical categories that did not explain at least 5% of any of the phenotype group were regrouped under the "Other" category. In each group, anatomical category representing less than 1% were approximated to 0. a) results obtained with variants detected with univariate test and b) results with variants detected with multivariate tests.
(TIF)

**S23 Fig. Clustering criterion by number of clusters and phenotype set.** In all panels, the x-axis is the number of clusters derived using the union of significant SNPs in the *Omnibus*, *SumZ*$_\mathrm{genet}$ and univariate tests. On the left column, the y-axis represents the BIC (Bayesian Information Criteria). On the right column, the y-axis is the Silhouette criteria (see Methods). Each line corresponds to a different group of phenotypes.
(TIF)

**S24 Fig. Clustering for the *Metabolism* set using SNPs detected by univariate analysis only.** The x-axis is the number of clusters derived using the significant SNPs univariate tests. On the left column, the y-axis represents the BIC (Bayesian Information Criteria). On the right column, the y-axis is the Silhouette criteria (see Methods).
(TIF)

**S25 Fig. Alluvial plot and heatmaps for the Anthropometry GWAS set.** On the left panel, the alluvial plot represents the re-assignment of SNPs from univariate analysis to clusters. To emphasize the relative genetic contribution to phenotypes, SNPs from each phenotype block were weighted by their variance explained to that phenotype. On the right panel, the heatmap represents the multi-trait signatures. Each line is a SNPs, each column, a trait. The gradient of color represents the strength of the Z-scores.
(TIF)

**S26 Fig. Alluvial plot and heatmap for the Cardiovascular GWAS set.** On the left panel, the alluvial plot represents the re-assignment of SNPs from univariate analysis to clusters. To emphasize the relative genetic contribution to phenotypes, SNPs from each phenotype block were weighted by their variance explained to that phenotype. On the right panel, the heatmap represents the multi-trait signatures. Each line is a SNP, each column, a trait. The gradient of color represents the strength of the Z-scores.
(TIF)

**S27 Fig. Alluvial plot and heatmap for the Immunity GWAS set.** On the left panel, the alluvial plot represents the re-assignment of SNPs from univariate analysis to clusters. To emphasize the relative genetic contribution to phenotypes, SNPs from each phenotype block were weighted by their variance explained to that phenotype. On the right panel, the heatmap represents the multi-trait signatures. Each line is a SNP, each column, a trait. The gradient of color represents the strength of the Z-scores.
(TIF)

**S28 Fig. Alluvial plot and heatmap for the Composite GWAS set.** On the left panel, the alluvial plot represents the re-assignment of SNPs from univariate analysis to clusters. To emphasize the relative genetic contribution to phenotypes, SNPs from each phenotype block were weighted by their variance explained to that phenotype. On the right panel, the heatmap represents the multi-trait signatures. Each line is a SNP, each column, a trait. The gradient of color represents the strength of the Z-scores.
(TIF)

**S29 Fig. Alluvial plot and heatmap for the Psychiatric GWAS set.** On the left panel, the alluvial plot represents the re-assignment of SNPs from univariate analysis to clusters. To emphasize the relative genetic contribution to phenotypes, SNPs from each phenotype block were weighted by their variance explained to that phenotype. On the right panel, the heatmap represents the multi-trait signatures. Each line is a SNP, each column, a trait. The gradient of color represents the strength of the Z-scores.
(TIF)

**S30 Fig. Alluvial plot and heatmap for the Metabolism GWAS set.** On the left panel, the alluvial plot represents the re-assignment of SNPs from univariate analysis to clusters. To emphasize the relative genetic contribution to phenotypes, SNPs from each phenotype block were weighted by their variance explained to that phenotype. On the right panel, the heatmap represents the multi-trait signatures. Each line is a SNP, each column, a trait. The gradient of color represents the strength of the Z-scores.
(TIF)

**S31 Fig. Alluvial plot and heatmap for all GWAS combined.** On the left panel, the alluvial plot represents the re-assignment of SNPs from univariate analysis to clusters. To emphasize the relative genetic contribution to phenotypes, SNPs from each phenotype block were weighted by their variance explained to that phenotype. On the right panel, the heatmap represents the multi-trait signatures. Each line is a SNP, each column, a trait. The gradient of color represents the strength of the Z-scores.
(TIF)

**S32 Fig. Heatmap for the *Metabolism* SNPs for cardiovascular traits and Alzheimer disease.** the heatmap represents the impact multi-trait signatures detected on the METABOLISM set on disease that have been linked to hyperlipidemia and diabetes. Each line is a SNP, each column, a trait. The gradient of color represents the strength of the Z-scores.
(TIF)

**S1 Table. GWAS considered.**
(XLSX)

**S2 Table. Variance-covariance matrix of Z-score under the null hypothesis.**
(XLSX)

**S3 Table. Genetic correlation.**
(XLSX)

**S4 Table. New associations for the Metabolism phenotype set.**
(XLSX)

**S5 Table. New associations for the Cardiovascular phenotype set.**
(XLSX)

**S6 Table. New associations for the Immunity phenotype set.**
(XLSX)

**S7 Table. New associations for the Anthropometry phenotype set.**
(XLSX)

**S8 Table. New associations for the Psychiatric phenotype set.**
(XLSX)

**S9 Table. New associations for the Composite phenotype set.**
(XLSX)

**S10 Table. New associations for the all the phenotypes considered together.**
(XLSX)

**S11 Table. Significant tissue enrichment by joint test and by phenotype set.**
(XLSX)

**S12 Table. Significant tissue enrichment for univariate GWAS derived with FUN-LDA.**
(XLSX)

**S13 Table. Significant tissue enrichment per multitrait association profile derived with FUN-LDA.**
(XLSX)

**S14 Table. Significant Go term enrichment for all phenotype sets and multitrait association profiles with FUMA.**
(XLSX)

**S15 Table. Significant KEGG term enrichment by phenotype sets and multitrait association profiles with FUMA.**
(XLSX)

**S16 Table. Significant SNPs for the Metabolism set that were retained in final clusters.**
(XLSX)

**S17 Table. Significant SNPs for the Immunity set that were retained in final clusters.**
(XLSX)

**S18 Table. Significant SNPs for the METABOLISM set retained in final clusters and with negligible linkage disequilibrium (R2 < 0.05).**
(XLSX)

**S19 Table. Sign test for z-score of diseases phenotype associated with metabolic traits.**
(XLSX)

**S20 Table. Potential drug targets mapped to multi-trait profiles for the Metabolism, Immunity, Cardiovascular phenotype set.**
(XLSX)

**S1 Data. FUMA parameter for functional analysis.**
(ZIP)

## Acknowledgments

This research has been conducted using the UK Biobank Resource under Application Number 42260.

## Author Contributions

**Conceptualization:** Hanna Julienne, Bjarni J. Vilhjálmsson, Hugues Aschard.

**Data curation:** Hanna Julienne, Amaury Vaysse.

**Formal analysis:** Hanna Julienne, Vincent Laville, Zihuai He, Carla Lasry, Andrey Ziyatdinov, Bjarni J. Vilhjálmsson, Hugues Aschard.

**Methodology:** Hanna Julienne.

**Software:** Hanna Julienne, Zachary R. McCaw, Vincent Guillemot, Cyril Nerin, Pierre Lechat, Hervé Ménager.

**Supervision:** Hugues Aschard.

**Writing – original draft:** Hanna Julienne, Hugues Aschard.

**Writing – review & editing:** Vincent Laville, Zachary R. McCaw, Wilfried Le Goff, Marie-Pierre Dube, Peter Kraft, Iuliana Ionita-Laza, Bjarni J. Vilhjálmsson, Hugues Aschard.

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
