## [Decision Letter · Decision Letter 0]

29 Mar 2021

Dear Dr Julienne,

Thank you very much for submitting your Research Article entitled 'Multitrait GWAS to connect disease variants and biological mechanisms' to PLOS Genetics.

The manuscript was fully evaluated at the editorial level and by independent peer reviewers. The reviewers appreciated the attention to an important topic but identified some concerns that we ask you address in a revised manuscript

We therefore ask you to modify the manuscript according to the review recommendations. Your revisions should address the specific points made by each reviewer.

[LINK]

Yours sincerely,

Ron Do

Guest Editor

PLOS Genetics

Scott Williams

Section Editor: Natural Variation

PLOS Genetics

Please address all reviewers' comments including Reviewer 1's comments on clarifying whether the biological and clinical insights obtained from this work are meant to be suggestive or whether the authors believe they are strong enough on its own to guide future action / decision making.

Reviewer's Responses to Questions

**Comments to the Authors:**

Reviewer #1: This paper includes a lot of analyses of summary-statistic GWAS results for multiple traits, and partitions SNP associations into clusters. There are interesting methodological details, and fairly detailed analyses and interpretations of several sets of phenotypes, with extensive supplemental material. The manuscript is written to focus on the phenotypic results with the methodological work largely in the supplement. It might be better to include more of the methods details in the main text.

The Introduction is not well written and needs more clarity and rationale. In fact the paper is comparing a set of multivariate methods, including at least one new method, and evaluating the performance and the identified results. But this message does not come through clearly at all in the current introduction.

Also, the introduction has a lot of space denoted to methods & results. I know this is a common practice in many articles, but in this case, the sentences’ meaning is unclear. The introduction should spend more time on the motivation. For example:

“First, characterizing and comparing the relative performances of alternative multitrait association models, we found strong specificity of the signal identified by each approach,…”

These “alternative…models” are not mentioned. Different kinds of models? Different assumptions? Why where they chosen? Then you say “….identified by each approach” but we don’t know what the approaches are.

Results

Performance of the different multivariate methods are discussed in the main text – when they agree, when they disagree, what kinds of features they agree on. But it is done superficially rather than exploring the specific assumptions behind each model. Presumably, a more in-depth comparison can be achieved through the simulation studies, but none of this appears in the main text. For example, are each SNP effects on phenotypes correlated? Or are the phenotypes correlated because the SNP acts on a shared latent factor that increases risk of each phenotype? Or are the phenotype correlated due to residual (non-genetic) correlations?

Many interesting findings are buried in one sentence: e.g. “We also developed corrections for several critical real data issues related to model misspecification (Figs S7 to S12) and missing data (Fig.S13)”. Some of these results should be in the main body of the paper. Similarly, in Methods, you say briefly “we implemented additional tools to estimate the per SNPs sample size…”. In the relevant supplemental material, the formulas and Figures relating to sample size are shown without sufficient explanation. It is difficult to decipher what is the problem being addressed in these supplemental sample size sections.

Existing multivariate methods were cited, but not used. Is there a reason for this?

Clustering: I am sure the authors are well aware of the challenges involved in interpreting clusters. Here, only SNPs that showed significance at 10-8 with at least one method are included in the clustering. Bootstrapping was used to look at cluster stability w.r.t. the number of clusters and the BIC/Silhouette criteria, but these would all be different if another threshold were used. Some clusters would have to be found given the thresholding effects. In fact, the authors acknowledge (in some way) the ad hoc nature of their analyses when they say:

“These distinct multitrait association profiles might arise because their variants belong to distinct genetic functional groups. Understanding whether those genetic functional groups are only statistical construction or correspond to meaningful biologically mechanism is critical. In the latter, it means that data-driven approach, such as the one proposed in the present study, can be used to dissect the genetic contribution of many complex human phenotypes.”

But this crucial text is in the middle of results, whereas I think this needs to be highlighted in the introduction.

An important message that needs to come through better is whether the conclusions that can be obtained through this kind of multivariate analysis of summary GWAS data is comparable enough to multivariate analyses of individual-level data to enable making decisions about potential drugs or treatments. Or is it just suggestive.

Could some UKbiobank data be used to do some structural equations modelling including individual level data, and to confirm some of the clustering relationships seen? i.e. could directed graphical relationships be estimated in the individual level data?

Minor

Careful editing would be helpful to clean up prepositions and articles, as well as general spelling and grammar here and there. A few examples are provided here:

• “relevance of multitrait association tests, there have” : probably should be “they”

• “We performed a series of analyses”

• “To understand further the relative performance of those three tests (omnibus, sumZica, sumZg) along with the univariate test”

And general wording would also benefit from good editing. For example:

• “we explored which multitrait signal was associated with the largest increase in detection per test. For that aim,…” The use of “For that aim,…” here is awkward.

• “…median chi-squared were elevated for the any …” should be “median chi-squared tests were elevated for the any…

Supplement, section on “Theoretical comparison with MANOVA”:

• K>>N should be K<<n

Clustering: Fig S22 should indicate the number of clusters chosen.</n

Reviewer #2: This study tackles the problem of finding pleiotropic loci that affects multiple traits (multi-trait analysis) and interpreting the complex genetic effect patterns. This is an important problem in the current genetics field, I think. The paper is extremely well written and most of the parts can be easily understood. The analyses and figures are fascinating.

I only have a few minor comments.

The authors are quite modest and do not argue that they developed these methods: the various sum of squares and omnibus. (They all look quite straightforward, and leave no room for controversy) Although omnibus is quite widely used, I think there’s some novelty in using PCs (driven from various sources) as weights for FE-meta. It’s a pretty simple idea, but it’s important to note that it’s certainly not what typical geneticist can bring up from his/her head instantly. I mean, it’s a good idea. I couldn’t find any citation there? So, if using PC as weight direction is what the authors have developed, I think it’s OK to advertise it as it is. (including the independent component analysis part)

A bit of discussion about possible multiple testing correction issue for applying combination of Z-r, Z-ica, and omnibus together would be beneficial.

Although details of GMM / MGMM are not described, only citing biorxiv paper, it should be better to reiterate how these frameworks work in a brief version here, so that this paper can be self-contained.

I would be really interested if the code that can automatically perform (1) some kinds of standard preprocessing including summary imputation, (2) various tests, and (3) visualization of Figure 2, 3, and 4. (Or generation of input formatted-data for Figure 2,3,4 along with the standard script). That will really help the community.

Reviewer #3: Review: Multitrait GWAS to connect disease variants and biological mechanisms

Overview

Julienne et al integrate GWAS summary statistics from multiple phenotypes to provide insights into shared genetic architecture and biological mechanisms. They leverage previously described multi-trait association methods to identify GWAS signals shared across 36 phenotypes and apply a novel clustering approach to group underlying association signals into broad biological categories. They highlight results from immune- and metabolism-related phenotypes to shed light on shared pathways. The methods and analyses are rigorous, and the takeaways should be of broad interest to the statistical genetics community. I found the manuscript to be well written, underlying assumptions thoroughly tested with simulations, and results to be generally well supported. With that said, I have a few comments.

Major Comments

1. Integrating multiple GWAS data from varying cohorts and studies requires great care in handling allele coding, varying sample size, and missingness. The authors have clearly spent a good deal of time considering the practical challenges and I was impressed with their systematic approach to each issue with rigorous statistical modeling and accompanying simulations. Similarly, I appreciated the thorough step-by-step derivations in the supplementary material. It was clear this was a considerable effort on part of the authors and sets a standard for how supplementary methods should be presented and for that they should be commended.

2. The authors note the need to prune SNPs that lacked a clear cluster assignment using large values of entropy as a metric. Can the authors perform analyses to provide some indication as to why these SNPs were unable to be assigned to any cluster with high confidence? Is this the result of similar functionality across clusters [and thus the method cannot assign to any with certainty], or is it more likely due to non-shared functionality and lack of a representative cluster?

3. With clusters acting as proxies for biological pathways, it would be interesting to see if tissue enrichment varies across clusters for a specific phenotype group.

Minor Comments

1. Figure 3 caption has a formatting or word-to-pdf conversion error, “ -cell function”.

**Have all data underlying the figures and results presented in the manuscript been provided?**

Reviewer #1: Yes

Reviewer #2: Yes

Reviewer #3: Yes

PLOS authors have the option to publish the peer review history of their article (what does this mean?). If published, this will include your full peer review and any attached files.

Reviewer #1: No

Reviewer #2: No

Reviewer #3: No

---

## [Decision Letter · Decision Letter 1]

12 Jul 2021

Dear Dr Julienne,

We are pleased to inform you that your manuscript entitled "Multitrait GWAS to connect disease variants and biological mechanisms" has been editorially accepted for publication in PLOS Genetics. Congratulations!

Yours sincerely,

Ron Do

Guest Editor

PLOS Genetics

Scott Williams

Section Editor: Natural Variation

PLOS Genetics

Comments from the reviewers (if applicable):

Reviewer's Responses to Questions

**Comments to the Authors:**

Reviewer #1: The authors have responded adequately to all comments.

Reviewer #2: The authors have successfully addressed my comments and I don't have further comments. I hope that the implementation will be used widely.

Reviewer #3: The authors have addressed all of my initial comments.

**Have all data underlying the figures and results presented in the manuscript been provided?**

Reviewer #1: None

Reviewer #2: Yes

Reviewer #3: Yes

PLOS authors have the option to publish the peer review history of their article (what does this mean?). If published, this will include your full peer review and any attached files.

Reviewer #1: No

Reviewer #2: No

Reviewer #3: No

**Data Deposition**

http://datadryad.org/submit?journalID=pgenetics&manu=PGENETICS-D-21-00084R1

**Press Queries**

---

## [Editor Report · Acceptance letter]

24 Aug 2021

PGENETICS-D-21-00084R1 

Multitrait GWAS to connect disease variants and biological mechanisms 

Dear Dr Julienne, 

We are pleased to inform you that your manuscript entitled "Multitrait GWAS to connect disease variants and biological mechanisms" has been formally accepted for publication in PLOS Genetics! Your manuscript is now with our production department and you will be notified of the publication date in due course.

With kind regards,

Andrea Szabo

PLOS Genetics

On behalf of:
